# A geographically weighted random forest approach for evaluate forest change drivers in the Northern Ecuadorian Amazon

**Fabián Santos**[1]*, **Valerie Graw**[2], **Santiago Bonilla**[1,3]

**1** Research Center for the Territory and Sustainable Habitat, Universidad Tecnológica Indoamérica, Quito, Ecuador, **2** Center of Remote Sensing of Land Surfaces (ZFL), University of Bonn, Bonn, Germany, **3** Departament of Forest Engineering. E.T.S.I.A.M., Campus de Excelencia Internacional Agroalimentario (ceiA3), Universidad de Córdoba, Córdoba, Spain

\* fabian_santos_@hotmail.com

**Data Availability Statement:** All data from MAE are available from: http://suia.ambiente.gob.ec/web/suia/anexos-nivel-referencia All data from IGM are available from: http://www.geoportaligm.gob.ec/portal/index.php/cartografia-de-libre-acceso-

## Abstract

The Tropical Andes region includes biodiversity hotspots of high conservation priority whose management strategies depend on the analysis of forest dynamics drivers (FDDs). These depend on complex social and ecological interactions that manifest on different space–time scales and are commonly evaluated through regression analysis of multivariate datasets. However, processing such datasets is challenging, especially when time series are used and inconsistencies in data collection complicate their integration. Moreover, regression analysis in FDD characterization has been criticized for failing to capture spatial variability; therefore, alternatives such as geographically weighted regression (GWR) have been proposed, but their sensitivity to multicollinearity has not yet been solved. In this scenario, we present an innovative methodology that combines techniques to: **1)** derive remote sensing time series products; **2)** improve census processing with dasymetric mapping; **3)** combine GWR and random forest (RF) to derive local variables importance; and **4)** report results based in a clustering and hypothesis testing. We applied this methodology in the northwestern Ecuadorian Amazon, a highly heterogeneous region characterized by different active fronts of deforestation and reforestation, within the time period 2000–2010. Our objective was to identify linkages between these processes and validate the potential of the proposed methodology. Our findings indicate that land-use intensity proxies can be extracted from remote sensing time series, while intercensal analysis can be facilitated by calculating population density maps. Moreover, our implementation of GWR with RF achieved accurate predictions above the 74% using the out-of-bag samples, demonstrating that derived RF features can be used to construct hypothesis and discuss forest change drivers with more detailed information. In the other hand, our analysis revealed contrasting effects between deforestation and reforestation for variables related to suitability to agriculture and accessibility to its facilities, which is also reflected according patch size, land cover and population dynamics patterns. This approach demonstrates the benefits of integrating remote sensing–derived products and socioeconomic data to understand coupled socioecological systems more from a local than a global scale.

escala-50k/ All data from Natural Earth are available from: https://www.naturalearthdata.com/ All census data from INEC are available from: https://www.ecuadorencifras.gob.ec/institucional/home/

**Funding:** SENESCYT: data collection and analysis. Universidad Indoamérica: decision to publish and preparation of the manuscript.

**Competing interests:** The authors have declared that no competing interests exist.

## 1.1 Introduction

The Tropical Andes is a mountainous region at the base of the Andes ridge. Due to its altitudinal gradient, it is characterized by 23 ecoregions and 8 bioregions [1], and it provides important economic and ecological services to almost 40 million inhabitants [2]. The region is recognized as an endangered biodiversity hotspot of high conservation priority [3,4], where population growth and agriculture expansion [5,6] are the major driving forces of deforestation, contributing to potential impacts of climate change [7]. On the other hand, large-scale reforestation has been detected in some areas of Latin America [8], especially along old colonization fronts [9]. However, these areas are less studied or understood, and their role in forest recovery and restoration of important environmental services is ignored [10,11]. Therefore, analyzing forest dynamics drivers (FDDs), i.e., deforestation and reforestation, in the Tropical Andes is very important for conservation, climate change adaptation, and sustainability. This knowledge is decisive for countries like Ecuador, where most of the remaining native forests are located and deforestation rates have been the highest in South America for some years [12,13].

Forest dynamics are shaped by complex societal and ecological interactions, or drivers. Geist and Lambin [14] proposed a conceptual framework to facilitate the understanding of these drivers of land dynamics, classifying them as follows:

1. proximate causes (local level, direct agents);

2. underlying causes (different levels, socioeconomic processes); and

3. other causes (determined by environmental factors and social triggering events).

These drivers have been accepted by countries participating in Reducing Emissions from Deforestation and Forest Degradation (REDD+), but recent research has recognized that underlying causes are less frequently analyzed in Latin America [6,15]. Proximate causes are mostly identified through remote sensing–based techniques [16], while underlying causes can be more complex, as they rely on socioeconomic data. These data are frequently not available or reliable at the scale needed [17]. Moreover, impacts of globalization [18] and economic development [11,19] generate more complex scenarios.

In Ecuador, previous studies combined remote sensing products and socioeconomic data to identify FDDs. For instance, Southgate et al. [20] analyzed thematic cartography and census data in a regression analysis to highlight agricultural rents, spontaneous settlements, and land tenure insecurity as deforestation drivers in eastern Ecuador. Following a similar approach but adding survey data, Rudel et al. [9] discussed reforestation drivers observed among ethnic groups and their relationships between land-use practices, cultural backgrounds, and distance to roads in southern Ecuador. Later, Mena et al. [21] combined thematic cartography, census, and survey data in a spatial regression model to conclude that road accessibility and population density were the most important deforestation drivers in northern Ecuador. Similarly, Walsh et al. [22] identified that reforestation drivers were motivated by land security and distance to roads. More recently, Bonilla-Bedoya et al. [23] related deforestation processes to legal timber harvesting, road expansion, and poverty indices. From these studies, it can be observed that deforestation is not commonly associated with reforestation. In this paper, the evaluation of contrasting driving forces (e.g., population growth/decay, agricultural expansion/contraction) with regard to possible linkages defines the first research interest.

Processing of multivariate data for FDD analysis has made significant progress in recent years. For instance, advances in remote sensing and open access to satellite archives [24] have contributed to a better understanding of global land-cover and land-use changes [25]. As a

result, products derived from time series (e.g., spectral trends, class-level metrics) has been increasingly applied to explain driving forces, making it possible to identify direct drivers [26] or better understand ecosystem fragmentation [27]. Furthermore, censuses are common sources of socioeconomic data, while processing of these data is not common in FDD analysis of underlying causes. Obstacles such as boundary changes [28] and scale effects [29] are probably the most challenging, and different approaches have been proposed to solve them, including areal interpolation [30,31] and statistical modeling [32–34]. Among them, areal interpolation with dasymetric mapping is perhaps the most popular [35], as it can combine land-cover maps and census data to model population distribution more precisely than other methods [36]. Other advances are related to capturing the spatial variability of FDDs. In this regard, geographically weighted regression (GWR) [37] has been demonstrated to satisfy this objective [38,39], but is sensitive to local collinearity and can produce unreliable results [40]. Nonparametric algorithms such as random forest (RF) [41] have interesting applications for high-dimensional problems with correlated variables [42]. Moreover, implementation of RF with GWR has recently proposed [43] but further applications explaining variables relationships are yet to be evaluated. The design of an innovative methodology for analyzing FDDs using these techniques constitutes the second research interest of this paper.

As the Tropical Andes constitutes a complex mosaic of landscapes, a workflow to analyze its FDDs is presented in this paper. We conducted our research in the Northwestern Ecuadorian Amazon (NEA), a study area located in an altitudinal gradient that includes different bioregions and colonization fronts with heterogeneous socioeconomic settings. Our main objective was to explore a set of variable groups to observe how they influenced deforestation and reforestation rates in the NEA in 2000–2010. This period is known as the beginning of the dollarization and economic stabilization in Ecuador [44]. For this purpose, we designed and implemented an experimental methodology for FDD analysis that benefits from the novel techniques mentioned above. Two research questions guided our work:

- What are the theoretical and empirical implications of our experimental methodology in FDD analysis?

- What could be the linkages between the driving forces of deforestation and reforestation in the NEA during 2000–2010?

To answer these questions, we (i) explain how we calculated the forest change rates and time series–derived products, (ii) conduct dasymetric mapping for intercensal analysis, and (iii) briefly describe the variable groups before (iv) explaining our implementation of GWR and RF, together with a clustering and hypothesis test to summarize our results. The discussion considers the benefits and limitations of the proposed methodology and its contribution to the current knowledge of FDDs in the NEA.

## 1.2 Study area

The NEA covers 21,857 km$^2$ over an altitudinal gradient from 200 to 2,800 m.a.s.l. on the western slopes of the Andean Range (Fig 1). It includes 16 cantons (second-level administrative units in Ecuador), which are used in this research to identify specific zones in the NEA. According to Olson et al. [1], two ecoregions exist in the NEA: the Napo moist forests and the Eastern Cordillera real montane forest. The latter is of the highest conservation priority in Ecuador as it covers less than 33% of its original area [45]. Moreover, the NEA is characterized by extraordinary biodiversity, intense annual precipitation (1,500–4,500 mm), and a multitude of ecosystems [46]. Most of the soils are ferric, with low fertility and high aluminum toxicity, although volcanic and alluvial soils can be an exception [47,48]. Under these conditions, the

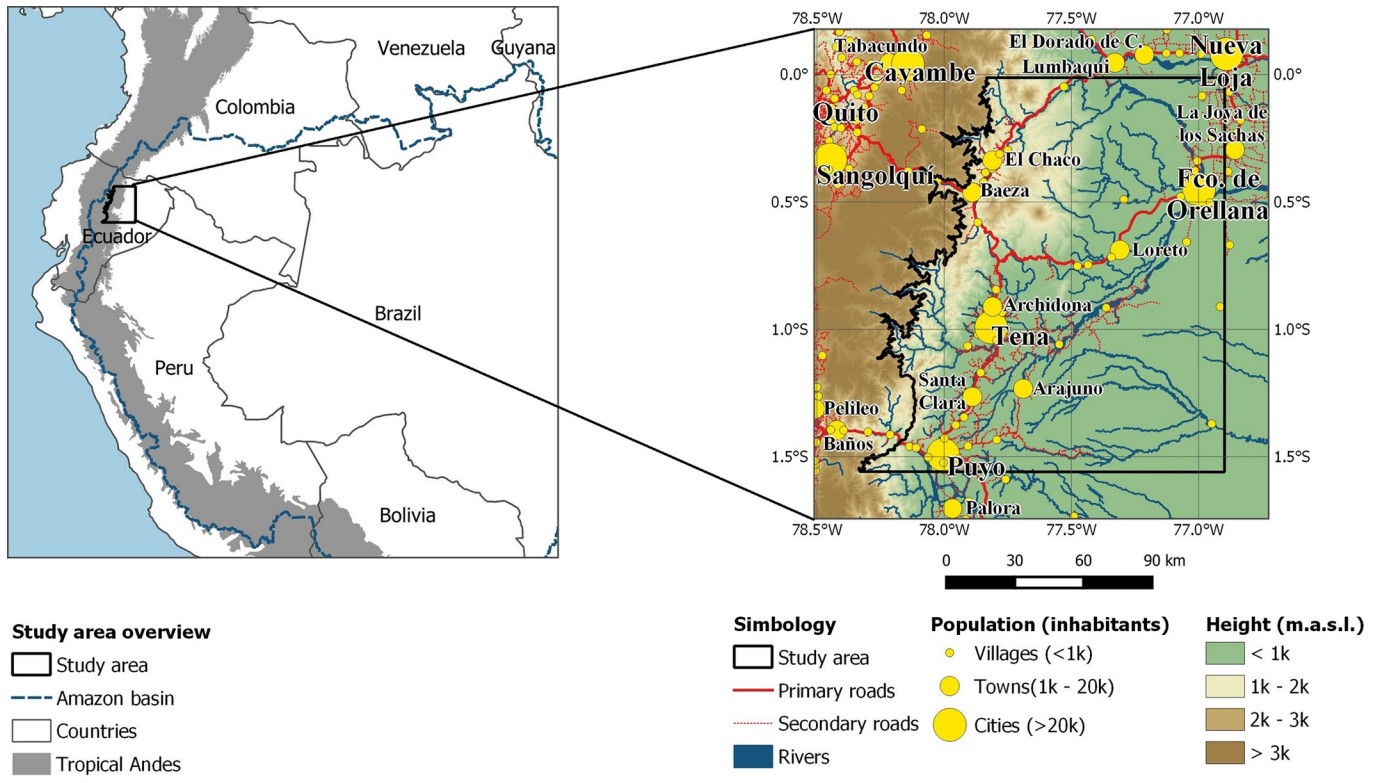

**Fig 1. Study area and its location in the Amazon basin.** Data from Natural Earth [52] and Instituto Geográfico Militar (IGM) [53].

agricultural limitations are well known; however, this does not prevent the native people from co-evolving with their natural environment [49]. Dramatic changes began in the 1970s with the exploration and extraction of oil, generating accelerated economic growth and industrialization [50]. Extensive road construction and the Agrarian and Colonization Reform of 1964 stimulated in-migration and rapid settlement over the whole Ecuadorian Amazon. According to Brown et al. [51], its population grew by 432% from 1950 to 1990, resulting in an urban system that followed the discovery of petroleum and the related economic opportunities. This led to disorganized and arbitrary colonization where land conflicts between the *colonos* (mestizo colonists) and native people were common and traditional land-use practices were replaced by extensive agriculture and cattle ranching [55].

Forest clearing in the Ecuadorian Amazon peaked during 1970–1990, when the deforestation rate was one of the highest in South America [12]. In the NEA, the forested areas experienced an 19.6% reduction (4130 km$^2$) by the end of 2014, principally due to pasture expansion for cattle ranching [56] (Fig 2). However, this was less intense than in the northeast NEA, where oil fields were located [57]. The declaration of protected areas, which accounted for 29% of the area and few oil discoveries [58], contributed to a reduced interest in colonization and to deforestation. Improved road connections between Quito and Nueva Loja and recent oil discoveries motivated further colonization of remote areas [59]. Despite this, reports indicate a drop of deforestation from 92,800 to 74,000 ha/year$^{-1}$ in Ecuador since 1990 [60].

Later, financial instability led to a crisis that ended with the dollarization of the Ecuadorian economy in September of 2000. A reduction in the inflation rate from 96 to 7% was seen as an important sign of economic stabilization for the period 2000–2014 [44]. As consequence, Ecuador experienced an unprecedented wave of emigration, especially between 2000 and 2007

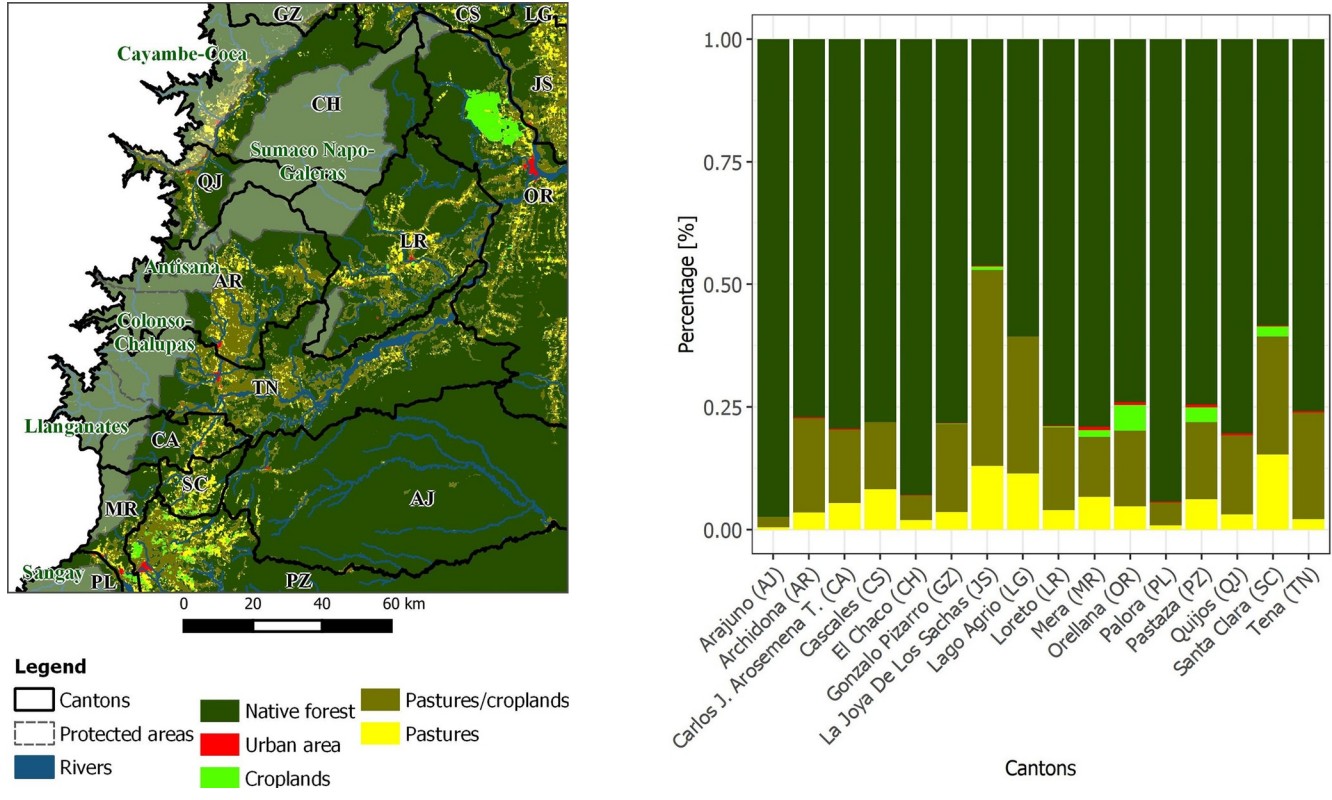

**Fig 2. Land cover for 2008 in the study area.** Data from Ministerio del Ambiente (MAE) [54].

(around 483.000 migrants) [61]. Nevertheless, the effects of migration and remittances through land-use change have been associated with an increase of agriculture activities rather than land abandonment and forest transition in Ecuador [62].

## 1.3 Methods

This research was fully implemented using R programming language [63] and integrating specific libraries for spatial data [64,65], database management [66], parallel processing [67], and data visualization [68]. For mapmaking, we used QGIS 3.4.3 Madeira [69]. Fig 3 shows the workflow of the proposed methodology.

### 1.3.1 Annual forest change rates and remote sensing time series–derived products

We collected a set of land-cover and land-cover change maps generated biannually for the period 1990–2014 in the NEA. They were generated for previous research to monitor long-term forest dynamics with scarce data [70], reporting an overall accuracy above 70%. Specifically, this approach uses the Landsat surface reflectance time-series product [71] to reduce it into cloud-free biennial composites. Then, it trains and executes a supervised-classification algorithm to derive land-cover maps, classified into 4 classes: evergreen forest, bamboo forest (*guadua* spp.), bare soil/infrastructure, and pasture/cropland. This collection of maps is post-processed and the classes are aggregated into forest and non-forest binaries to derive deforestation and reforestation areas. In the case of deforestation, the algorithm simply flags the date of conversion from forest to non-forest, while for reforestation it first considers a minimum time

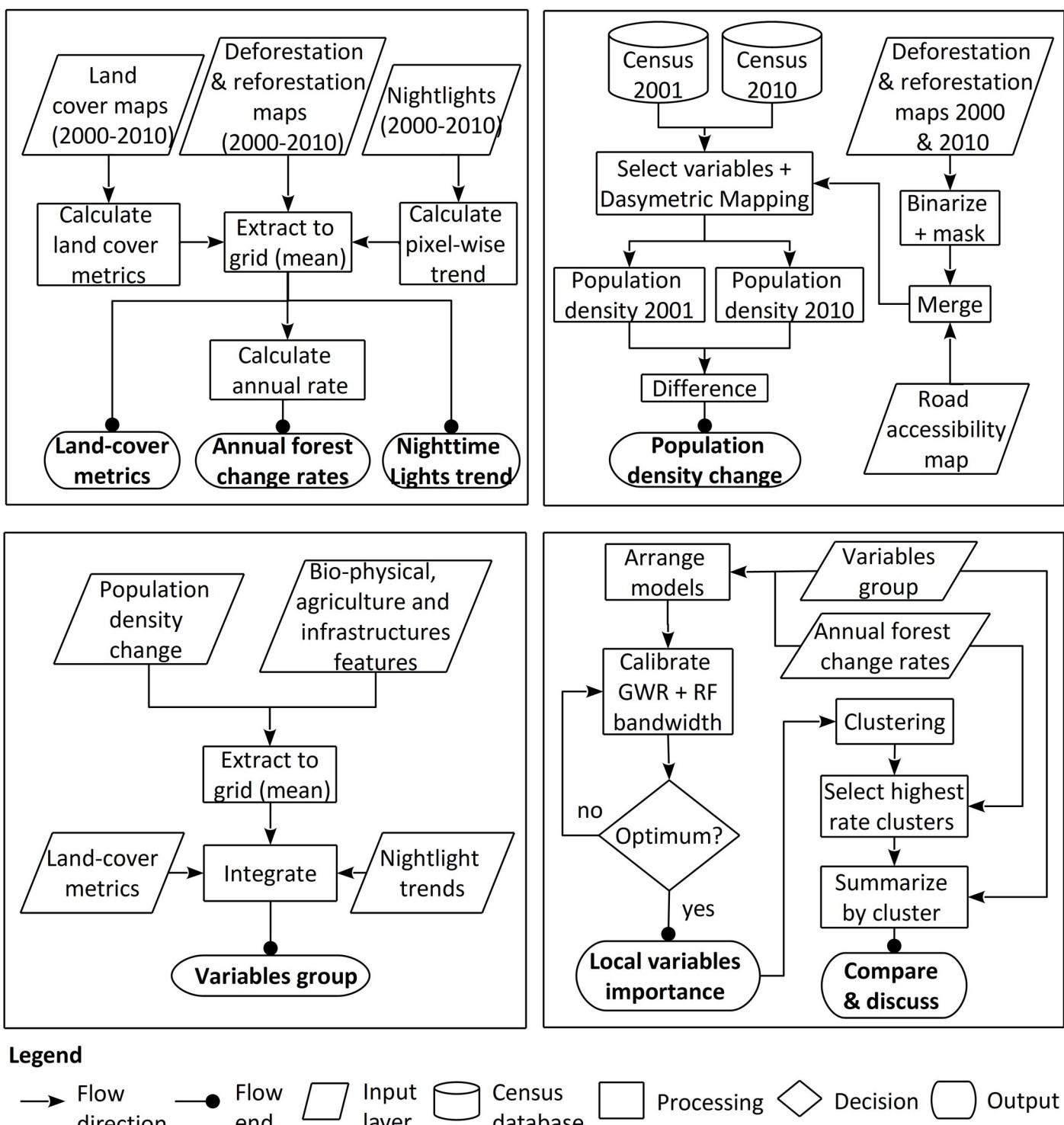

**Fig 3.** Flowchart of the methodology: (a) derivation of forest change rates and time-series products; (b) census processing with dasymetric mapping; (c) data integration; and (d) implementation of GWR and RF.

classified as forest after a disturbance (i.e., 4 years) to flag it as reforestation if the area remains as forest until the end of the time series. To derive the annual forest change rate, we first

removed areas less than 1 ha from deforestation and reforestation maps, as they frequently represent misclassified pixels [26]. Then, the annual forest change rate $q$ according to the UN Food and Agriculture Association (FAO) [72] was calculated, considering $A_1$ and $A_2$ as forest cover for time periods $t1$ and $t2$:

$$q = \left(\frac{A_2}{A_1}\right)^{1/(t2-t1)} - 1$$

To operate this, we created an analysis grid with a cell size of 400 ha and extracted for each cell the deforested and reforested areas between 2000 and 2010. These years were selected to match the census years used in this research (2001 and 2010) as well as the cell size to reduce processing time during calculation. The resulting deforestation and reforestation rates constituted the set of dependent variables analyzed (Fig 4), a summary of which is shown in Table 1.

Furthermore, historical land use influence ecological landscape functions and link cause–connection patterns [73]. For this, we derived land-cover frequencies of deforested and reforested areas to pasture/cropland and bare soil/infrastructure classes. This provided further information about LCLUC dynamics and helped us to determine the permanence or semipermanence of a specific land-cover class $Z$. For this, we stacked the land-cover maps used in deforestation and reforestation mapping to obtain time series $M_{1:n}$, which was split to match deforestation or reforestation date $i$. This gave us 2 segments, defining the conditions (1) after a change $M_a = M_{i:n}$ and (2) before a change $M_b = M_{1:(i-1)}$. For deforestation, the segment $M_a$ was used to determine the land-cover frequency $f$ of class $Z$ after a deforestation event $D_{fz}$. For

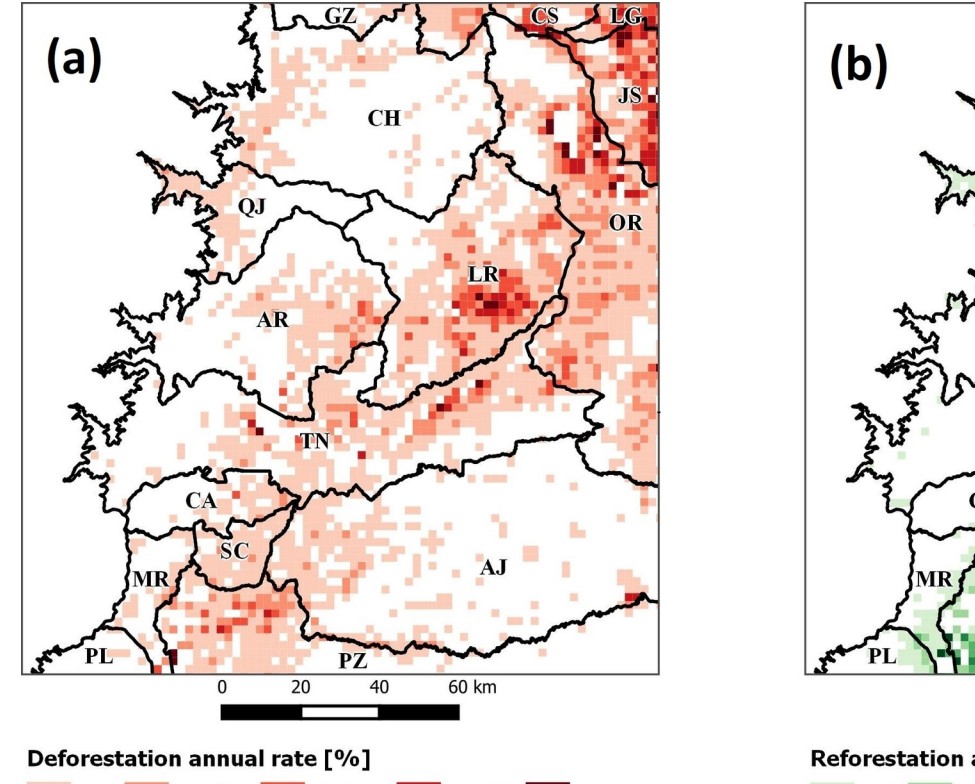
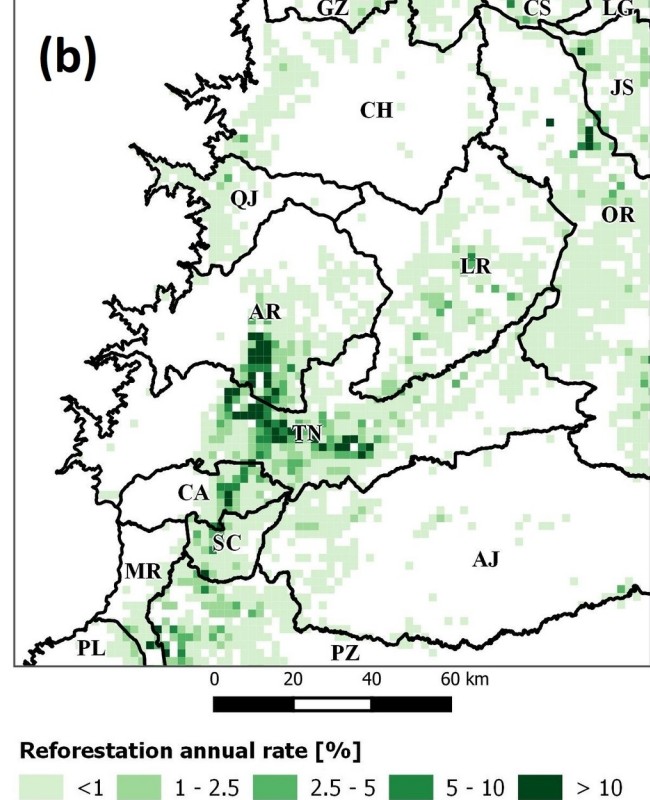

**Deforestation annual rate [%]**
 <1 1 - 2.5 2.5 - 5 5 - 10 > 10

**Reforestation annual rate [%]**
 <1 1 - 2.5 2.5 - 5 5 - 10 > 10

**Fig 4.** Annual forest change rates for: (a) deforestation and (b) reforestation. Data from Santos et al. (2018).

**Table 1. Descriptive statistics of dependent variables.**

| Variable | Prefix | q (%) | | Analysis grid (no. cells) | Total area (ha) | Data source |
|---|---|---|---|---|---|---|
| | | Mean | SD[2] | | | |
| Annual deforestation rate[1] | DEF | −1.29 | 6.02 | 2418 | 967,200 | [70] |
| Annual reforestation rate | REF | 2.08 | 11.23 | 1998 | 799,200 | |

[1] To facilitate map reading, absolute values from deforestation rates were used.

[2] SD, standard deviation.

this, we summed up the class occurrences in $M_a$ and divided the sum by the extent of the segment:

$$D_{fz} = \frac{|\{Z \in M_a\}| * 100}{|M_a|}$$

For reforestation, segment $M_b$ was used to determine the land-cover frequencies of class $Z$ before a reforestation event $R_{fz}$ happened. Similarly, it was calculated by adding up their occurrences in $M_b$ and dividing the sum by the extent of the segment:

$$R_{fz} = \frac{|\{Z \in M_b\}| * 100}{|M_b|}$$

These calculations gave us 4 layers in total, describing the land-cover frequencies for pasture/cropland and bare soil/infrastructure as percentages, for both deforestation (Fig 5A) and reforestation.

A final time series–derived product was obtained from the Visible Infrared Imaging Radiometer Suite (VIIRS) and its Nighttime Lights Time Series cloud-free composites (nightlights 2000–2010) [74]. This dataset constitutes a measure of visible and near-infrared emission sources at night (e.g., cities, towns, gas flares, and other sources of persistent lighting), which can refer to access to electricity and human development factors such as: access to education [75], emissions of $CO_2$ [76] or socioeconomic trends [77]. The latter motivated its use in this research to enrich socioeconomic parameters. For this, we used Google Earth Engine [78] to calculate a pixel-wise linear trend map for 2000–2010 using the stable light band from this dataset. From the resulting pixel-wise linear model, we extracted its slope to determine its trend (Fig 5B).

### 1.3.2 Dasymetric mapping and population change

We processed the 2001 and 2010 population censuses published by the National Institute of Statistics and Census of Ecuador (INEC) [79,80] to generate population density surfaces with dasymetric mapping. This technique allows redistribution of population counts from a set of areal units into a grid using land-cover maps. To implement it, we extracted the most detailed level of census information (census blocks) to avoid aggregation and bias effects [29]. Moreover, since rural population better explains conversion from forest to agricultural land [81], we used census blocks from rural areas with an average size of 4,995 ± 10,250 ha and filtered the population by working age (15–72 years). Following Mennis [82], we adapted his dasymetric mapping with areal weighting approach to operate with rural populations. We first binarized deforestation maps from 2000 and 2010 to produce 2 non-forest masks to represent areas where rural populations were mostly settled in a specific year. Since other features not related to rural population were also present in these masks (e.g., water bodies, cliffs, urban areas), we

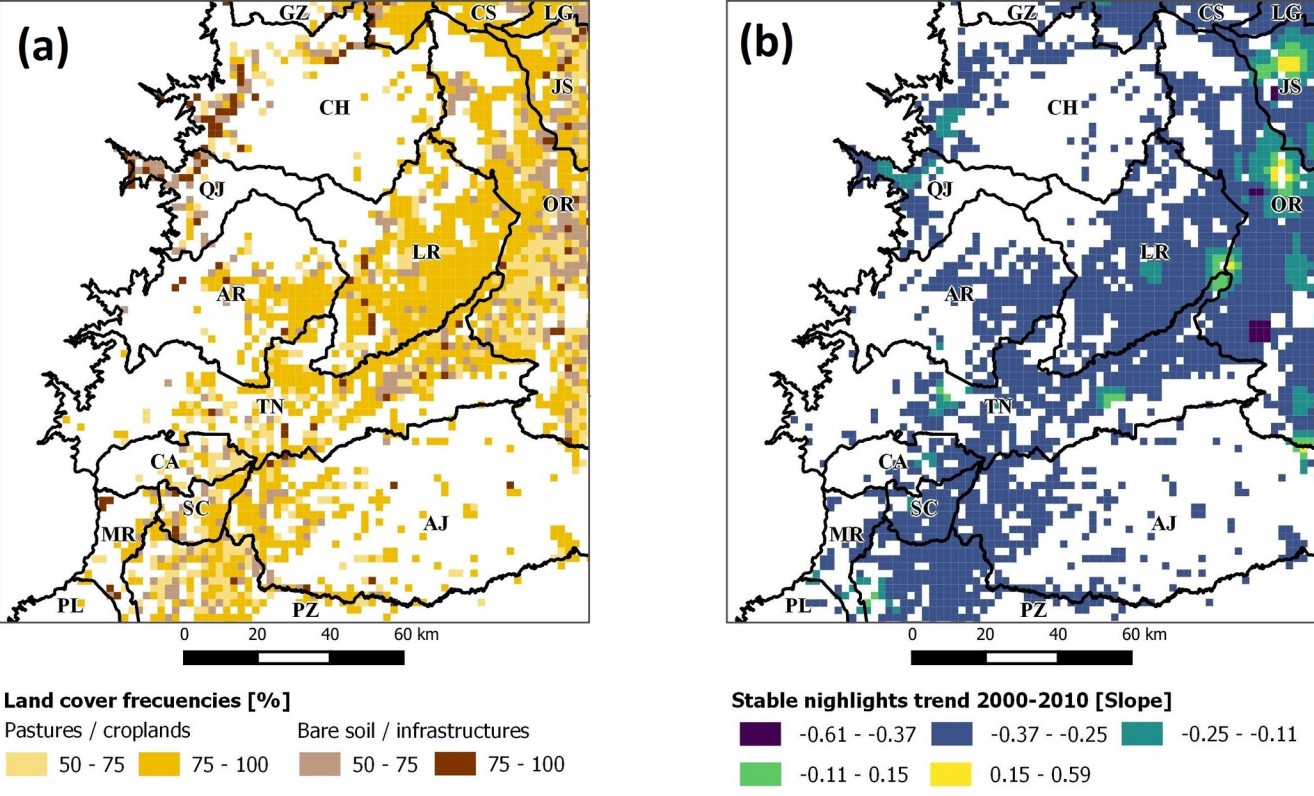

**Fig 5.** Time series–derived products for deforested areas: (a) land-cover frequencies and (b) stable nightlight changes.

identified and removed them in order to be consistent with our focus on rural populations. As further details were required to locate rural populations, a road accessibly model [83] was added to these masks to facilitate their identification. We assumed that higher rural population density would occur in areas with better road accessibility [21,84]; therefore, we reclassified the road accessibly map into 3 travel time ranges (high: 0–1 h; medium: 1–3 h; low: >3 h) to obtain what is called rural density classes. We considered only 3 classes, as the algorithm proposed by Mennis [82] was not tested with more than 3. In the next step, we identified census blocks that were almost completely within each rural density class to calculate their densities (Table 2).

**Table 2. Sampled census blocks and their population values.**

| Year | Locations of census blocks (cantons) | Rural density classes | Population (no. persons) | Area (ha) | Population density (persons/ha) | Sum density | Population density fraction |
|------|------|------|------|------|------|------|------|
| 2001 | LR[1] | High | 36 | 4293 | 0.008 | 0.021 | 0.394 |
|      | AJ[2] | Medium | 34 | 3195 | 0.010 | 0.021 | 0.501 |
|      | AR[3] | Low | 23 | 10420 | 0.002 | 0.021 | 0.103 |
| 2010 | LR | High | 68 | 4297 | 0.015 | 0.030 | 0.745 |
|      | AJ | Medium | 42 | 3269 | 0.012 | 0.030 | 0.604 |
|      | AR | Low | 18 | 10389 | 0.001 | 0.030 | 0.081 |

[1] Refer to Loreto canton.

[2] Refer to Arajuno canton.

[3] Refer to Archidona canton.

We then averaged their population density fraction to obtain the next values: 0.570, 0.553, and 0.092, which correspond to high, medium, and low rural density classes, respectively. These values are dimensionless and are obtained by:

$$d_{uc} = \frac{p_{uc}}{p_{hc} + p_{mc} + p_{lc}}$$

where $d_{uc}$ is the population density fraction of rural class $u$ in census block $c$, and $p_{uc}$ is the population density (persons/ha) of rural class $u$ in census block $c$ and is divided by the sum of all rural density classes (high $h$, medium $m$, and low $l$) and their census blocks. After calculating with all census blocks, the next step in the algorithm is to evaluate the area ratio. This operation divides the area of class $u$ by 33.3% to adjust densities equally according to the difference in area of each rural density class within that census block. This can be expressed as:

$$a_{ub} = \frac{n_{ub}/n_b}{0.33}$$

where $a_{ub}$ is the area ratio of rural class $u$ and $n_{ub}$ is the number of grid cells of rural class $u$ in census block $b$, and $n_b$ is the number of grid cells in census block $b$. The next step is to calculate the total fraction by multiplying $d_{uc}$ and $a_{ub}$ and dividing that result by the result of that same expression for all 3 rural classes ($h$, $m$, $l$) in that census block:

$$f_{ubc} = \frac{(d_{uc} * a_{ub})}{[(d_{hc} * a_{hb}) + (d_{mc} * a_{mb}) + (d_{lc} * a_{lb})]}$$

where $f_{ubc}$ is the total fraction of rural class $u$ in census block $b$ of spatial unit $c$. The final step in the algorithm is to assign each rural class to grid cells within that census block. This was done by dividing the population assigned to the rural class evenly among the grid cells in the census block that has that rural class. This can be expressed as:

$$pop_{ubc} = \frac{(f_{ubc} * pop_b)}{n_{ub}}$$

where $pop_{ubc}$ is the population assigned to one grid cell of rural class $u$ in census block $b$ of spatial unit $c$. The result is a population density surface (Fig 6), which represents the number of persons by pixel area (which was set as 1 ha to avoid census block elimination during vector-to-raster conversion). We iterated the algorithm with 20 census variables (see Section 1.3.3) to obtain population density surfaces. In all cases, the pycnophylactic property [85] was verified by adding population density surface pixels and comparing them with their original values. Only incomplete census blocks were observed as suspicious, because their geometry was modified during the study area extraction. Consequently, their counts were adjusted proportional to their original areas before calculation. In the final step, population density surfaces for 2001 and 2010 were subtracted for each variable to obtain their change.

### 1.3.3 Variable groups

With the processed censuses, we defined a set of variable groups related to demographic features and their change between 2001–2010. These were selected based on similar research [86,87] and included the categories: age composition (Age), literacy level (Education), gender distribution (Gender), household structure (Household), spoken languages (Language), and work sectors (Work). These constituted the Socioeconomic and Sociocultural macro levels (Table 3). Another set of variable groups was defined following similar research [6] to include biophysical (Biophysical) and land cover (Land Cover) features, constituting the Landscape

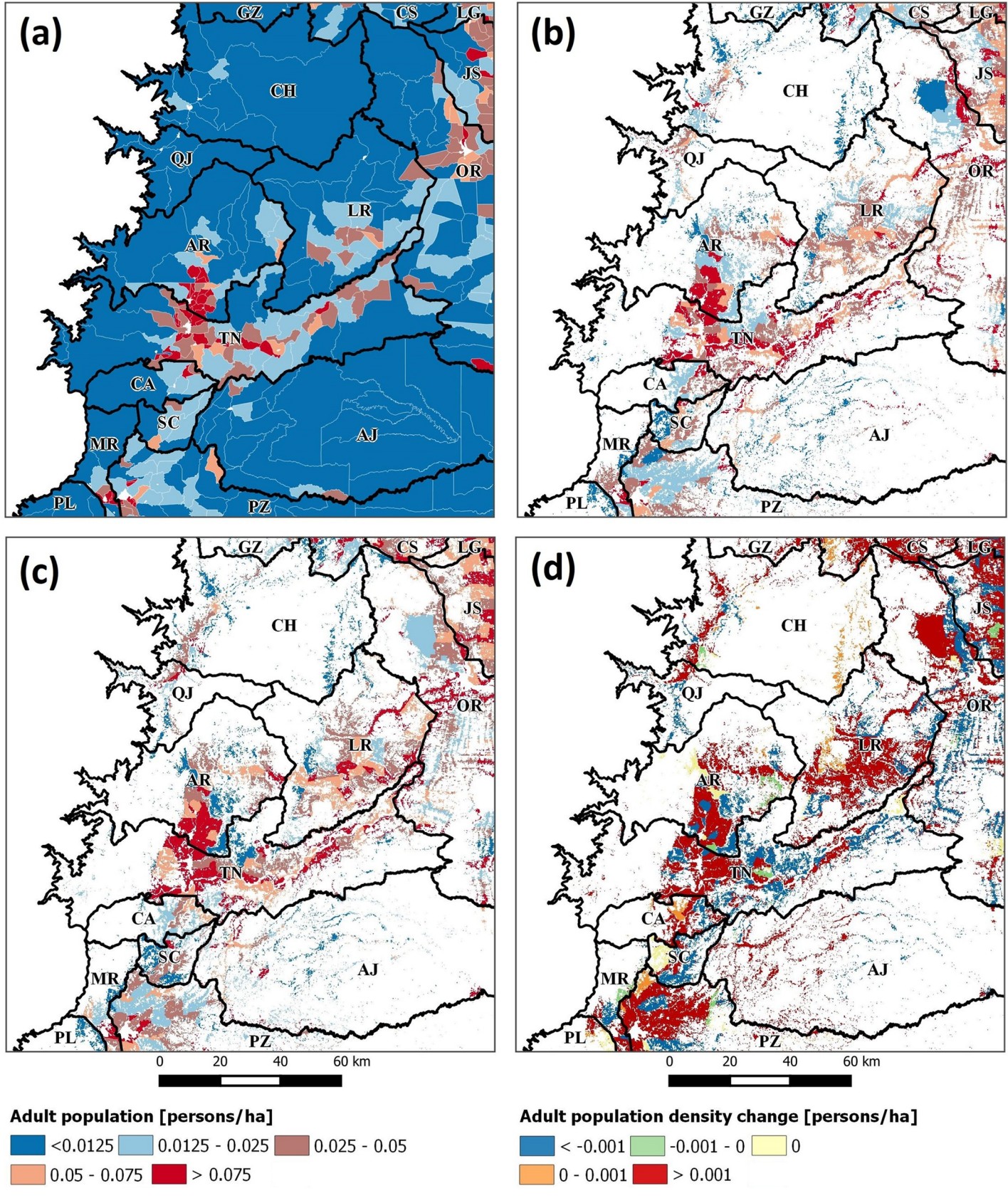

**Adult population [persons/ha]**

- <0.0125
- 0.0125 - 0.025
- 0.025 - 0.05
- 0.05 - 0.075
- > 0.075

**Adult population density change [persons/ha]**

- < -0.001
- -0.001 - 0
- 0
- 0 - 0.001
- > 0.001

**Fig 6.** Adult population density surfaces using: (a) census 2001, (b) dasymetric mapping 2001, and (c) dasymetric mapping 2010. The subtraction of the last two to derive adult population density change 2001–2010 is shown in (d).

macro-level (Table 4). Moreover, cost-distance models of agricultural collection facilities (palm oil, coffee, cacao, fruit, and milk products) were used as proxies of commercial agricultural activities (Agriculture). Another variable group (Infrastructures) considered the Euclidean distance to human-built infrastructures such as oil wells and mining blocks established between 2000 and 2010. Additionally, we include here the linear trend map from the nightlights 2001–2010 dataset. These variable groups formed the Commodities macro level (Table 4). A total of 34 variables were organized in 10 groups and 4 macro levels, with their values extracted into the analysis grid cells as averages.

### 1.3.4 Geographically weighted regression (GWR) and random forest (RF)

GWR is a statistical method to model spatial relationships under the assumption of spatial non-stationarity and location interdependency. It was conceived as an extension of linear regression analysis incorporating local estimates and surface representations of relationships among dependent and independent variables [37]. A GWR model can be specified as:

$$\gamma_i = \beta_{i0} + \sum_{k=1}^{m} \beta_{ik}\, \chi_{ki} + \varepsilon_i; \quad i = \{1, \ldots, n\}$$

where $\gamma_i$ is the dependent variable at $i$ location, $\beta_{i0}$ is the estimated intercept at $i$ location, $\chi_{ki}$ is a vector of $k = \{1,\ldots,m\}$ independent variables at $i$ location and $\varepsilon_i$ is the error term of the estimation at $i$ location. Since $i$ is considered as an $n \times n$ (with $n$ number of observations) diagonal matrix in GWR, its formulation for local parameter estimates at $i$ location is more conveniently expressed as:

$$\hat{\boldsymbol{\beta}}(i) = [\boldsymbol{X}^T \boldsymbol{W}(i) \boldsymbol{X}]^{-1} \boldsymbol{X}^T \boldsymbol{W}(i) \boldsymbol{\gamma}$$

where $\hat{\boldsymbol{\beta}}(i)$ is a vector of spatially weighted estimates for the $k$-th independent variables at $i$ location, $\boldsymbol{X}$ is a $n \times k$ matrix of independent variables, $\boldsymbol{W}(i)$ is the $n \times n$ diagonal weights matrix which ensure that observations near $i$ have the largest weight values rather than those further away [90], and $\boldsymbol{\gamma}$ is a vector of $k$ observations of the dependent variable. To define $\boldsymbol{W}(i)$ a weighting function is declared considering: (1) the type of distance between $i$ and its neighbors, (2) a kernel function specifying the weighting scheme, and (3) the bandwidth distance to control the number of observations within the kernel. Commonly, the Euclidean distance and the exponential kernel function are used as the weighting scheme [91]. The latter is defined by:

$$W_{ij} = \exp\left(-\frac{|d_{ij}|}{bw}\right)$$

where $W_{ij}$ is the weight assigned to observation $j$ for the estimation of $i$, $d_{ij}$ is the distance between $j$ and $i$, and $bw$ is the bandwidth. The latter defines GWR mapping sensibility, as large values result in global regression estimates, while small ones introduce randomness [92]. Moreover, $bw$ can be set as a fixed (constant distance) or an adaptive kernel (constant number of local observations). The latter is recommended, as it ensures a sufficient flow of information for each local calibration, while its size can be determined through cross-validation [93]. We implemented an adaptive exponential kernel function using Euclidean distance but tested different values for $bw = \{100,200,\ldots,800\}$ to determine it. Then, for each $i$ and its neighbors, we constructed an RF model instead of a linear regression.

**Table 3. Mean population densities change (2001–2010) for variables groups in socioeconomic and sociocultural macro levels using deforestation (*DEF*) and reforestation (*REF*) datasets.**

| Macro level | Variable group | Prefix | Variables[1] | *DEF* | *REF* | Data source |
|---|---|---|---|---|---|---|
| Socioeconomic | Age | D_ygr | Young population (15–25 y) | 0.006 | 0.005 | [79,80] |
| | | D_adt | Adult population (26–45 y) | 0.008 | 0.008 | |
| | | D_old | Older adult population (45–72 y) | 0.009 | 0.007 | |
| | Education | E_ilt | Illiterate | −0.001 | −0.002 | |
| | | E_pri | Primary education (1–6 y) | -0.0009 | -0.004 | |
| | | E_sec | Secondary education (7–12 y) | 0.018 | 0.020 | |
| | | E_hgr | Higher education (>13 y) | 0.003 | 0.004 | |
| | Work | W_agr | Agricultural workers | 0.010 | 0.008 | |
| | | W_ind | Industrial workers | −0.002 | −0.002 | |
| | | W_ser | Service workers | 0.0006 | 0.001 | |
| Sociocultural | Gender | G_chm | Chief male household | 0.008 | 0.007 | |
| | | G_pom | Male population | 0.012 | 0.010 | |
| | | G_chf | Chief female household | 0.002 | 0.002 | |
| | | G_pof | Female population | 0.012 | 0.011 | |
| | Household | H_sma | Small families (1–2 children) | 0.002 | 0.002 | |
| | | H_med | Medium families (3–5 children) | 0.004 | 0.004 | |
| | | H_lar | Large families (>5 children) | −0.014 | −0.016 | |
| | Language | L_spa | Speak Spanish[2] | 0.046 | 0.046 | |
| | | L_kcw | Speak *Kichwa*[3] | 0.029 | 0.032 | |
| | | L_wao | Speak *Huao Tededo*[4] | 0.0007 | -0.0002 | |

[1] All variables are reported as population densities (persons/ha).

[2] Most commonly spoken language by *colonos* in the NEA.

[3] Second most commonly spoken language and ethnicity in the NEA.

[4] The language of *Huaorani* people.

**Table 4. Mean values for variables groups in landscape and commodities macro levels using deforestation (*DEF*) and reforestation (*REF*) datasets.**

| Macro level | Variable group | Prefix | Variables and units | *DEF* | *REF* | Data source |
|---|---|---|---|---|---|---|
| Landscape | Biophysical | B_alt | Altitude (m.a.s.l.) | 656 | 716 | [88] |
| | | B_fer | Soil fertility (% organic matter) | 1–2 | 1–2 | |
| | | B_rfl | Annual rainfall (mm) | 3588 | 3658 | |
| | Land cover | C_bsl | Bare soil (% frequency) | 31 | 25 | [70] |
| | | C_pas | Pasture (% frequency) | 73 | 53 | |
| | | C_sze | Mean patch size (ha) | 6.5 | 4.9 | |
| | | C_fra | Fractal dimension index (unitless) | 1.07 | 0.79 | |
| Commodities | Agriculture | A_plm | Accessibility to oil palm extraction facilities (h)[1] | 1–3 | 1–3 | [83] |
| | | A_cao | Accessibility to coffee and cacao collection centers (h) [1] | 0.5–1 | 0.5–1 | |
| | | A_fru | Accessibility to fruit collection centers (h) [1] | 1–3 | 1–3 | |
| | | A_mlk | Accessibility to milk product collection centers (h) [1] | 1–3 | 1–3 | |
| | Infrastructure | I_oil | Distance to oil wells (perforated between 2000 and 2010) (m) [1] | 250 | 299 | [88,89] |
| | | I_min | Distance to mining blocks (assigned between 2000 and 2010) (m) [1] | 109 | 95 | |
| | | I_ngt | Stable nightlights trend 2000–2010 (slope) | −0.27 | −0.27 | |

[1] These variables were normalized (0–1) and inverted (i.e., maximum distance and travel time were assigned values near zero, contrary values near one) during GWR modelling to facilitate interpretation but were transformed to their original units for reporting.

RF is an ensemble learning method for classification and regression that produces multiple decision trees using bagging to select subsets of training samples and random feature selection to split them [41]. It is easy to compute and is tolerant to missing and multicollinear data [94], moreover it provides error estimates without requiring a validation dataset. During the training phase, it randomly sample with replacement, about two-third of the training samples (referred to as *in-bag* samples) for a given training set $T = \{1,\ldots,t\}$ to grow a specified number of trees to the largest extent possible, selecting randomly a number of variables $V = \{1,\ldots,v\}$ at each node to determine their split. This gives an ensemble of classification or regression trees, if $T$ and $V$ are bagged repeatedly $B$ times to grow trees with these samples. After training, it averages predictions from all individual regression trees or by taking the majority vote in classification. This can be summarized as:

For $b = \{1,\ldots,B\}$:

1. Sample randomly, with replacement, $n$ training samples and variables from $T$, $V$. Set them as $T_b$, $V_b$.

2. Train a classification or regression tree on $T_b$, $V_b$. Set it as $RF_b$.

   End for

   3. Average individual $RF_b$ results in regression or by taking the majority vote in classification and calculate model performance

To monitor error, the remaining one-third samples (referred to as *out-of-bag* samples or OBB) are used in an internal cross-validation technique [95], which computes the number of correct predictions. Other accuracy metrics are also possible to derive from OBB (e.g. Kappa, R-square, etc.). In addition, variables predictive power (or importance) can be calculated through different approaches (e.g. Gini index, accuracy decrease, permutation) but permutation is mostly recommended [96].

While the vast majority of RF problems can be solved with a unique (or global) model; here, we followed the approach of Georganos et al. [43] to combine GWR and RF to derive multiple spatially weighted (or local) RF models. This is possible if during the bagging step of RF, we assign to the neighbor observations of $i$ a sampling probability based on the distance weights or $W(i)$. For this, we can reshape step one from the previous RF workflow as follows:

For $b = \{1,\ldots,B\}$:

1. Apply $W(i)$ probabilities in sampling, with replacement, for $n$ training samples from $T$. Sample $n$ random variables from $V$. Set them as $T_b$, $V_b$.

   $\vdots$

Contrary to a global model, in this approach, an ensemble of spatially weighted (or local) RF models are obtained. Their features can be mapped and among them we can mention: **1)** local variables importance (*LVI*), which shows variables predictive power locally (or spatially), for each variable in $V$; **2)** prediction results, which can be also reported as probabilities; and **3)** model performance. The latter includes specific metrics for classification (e.g. kappa index, confusion matrix) or regression (e.g. r-squared, mean absolute deviation). To compute these features, in the next section we explain how we implemented GWR and RF.

### 1.3.5 Implementation of GWR and RF

To operate this GWR and RF as an algorithm, we used the ranger [42] and GWmodel [97] packages. The first case, is a fast C++ and R implementation of RF that allows weights for

sampling training observations. This parameter, called case weights in the ranger function, is used to define the spatial weights $W(i)$ for observations near $i$. The second case, is a complete toolbox for the geographically weighted approach, including functions for: regression analysis, spatial metrics, weight-decay functions, among others. Since the proposed algorithm (named from now as GWRF) involves multiple steps, we summarized them as a pseudocode:

```
Algorithm 1: Geographically weighted random forest (GWRF)
INPUTS
Sp: spatial dataset (with dependent and independent variables); Dep:
dependent variable name; K_fun: kernel function; K_typ: kernel type; K_bw:
kernel bandwidth;
OUTPUTS
LVI: variables importance; YHAT: prediction probabilities; ACC: accu-
racy metrics;
PROCEDURE
1: READ Sp; SET Dep as dependent variable; SET Outputs as an empty
list
FOR each i element IN Sp DO
    2: CALCULATE distances from all elements in Sp to i; SET them as D_i
    3: SORT D_i AND select those within K_bw; SET selected observations as
i_obs
    4: REMOVE variables with zero variance in i_obs
    IF Dep is categorical (classification problem)
        5: UPSAMPLE unbalanced classes in i_obs
    END IF
    6: CALCULATE spatial weights for i_obs applying the K_fun; SET it as W
(i)
    7: TRAIN Random Forest with i_obs applying W(i) as sampling probabil-
ities; SET it as RF_i
    8: EXTRACT LVI_i from RF_i; REMOVE variables with negative scores in
LVI_i
    IF number of variables in LVI_i are not equal to input Sp variables
number DO
        9: TRAIN Random Forest with i_obs applying W(i) as sampling prob-
abilities; UPDATE object RF_i
        10: EXTRACT LVI_i from RF_i; REMOVE variables with negative scores
in LVI_i
    END IF
    11: SET removed variables in LVI_i as zero
    12: EXTRACT predictions (probabilities, predicted value); SET it
as YHAT_i
    13: CALCULATE accuracy metrics (Kappa, R-squared, prediction fail-
ure, residual standard error); SET it as ACC_i
    14: SAVE LVI_i, YHAT_i, ACC_i into Outputs_i
    END FOR
15: MERGE Outputs_{i_1...i_n}; SAVE outputs as LVI, YHAT, ACC spatial datasets
END PROCEDURE
```

Note that the algorithm inputs require to define the dependent variable and we set it as *DEF* and *REF* to refer to the annual deforestation and reforestation rates (See Table 1). As the algorithm assume that the rest of variables are independent, we can express them according to their variables groups names:

$$Vars = \{Landscape, Commodities, Socioeconomic, Sociocultural\}$$

Where *Vars* refers to all variables described in Table 3 and Table 4. Now, we can represent the calibration of GWRF for *DEF* as:

$$GWRF(Sp = DEF|Vars, Dep = DEF, K_{fun} = exponential, K_{typ} = adaptative, K_{bw}$$
$$= \{100\ldots, 800\})$$

And similarly, for *REF* as:

$$GWRF(Sp = REF|Vars, Dep = REF, K_{fun} = exponential, K_{typ} = adaptative, K_{bw}$$
$$= \{100\ldots, 800\})$$

After reading and preparing models (step 1), a loop is defined for process each element in the spatial dataset *Sp*. This processing included operations such as:

- Data cleaning (steps 4, 5, 8, 10 and 11), following Genuer et al. [98] for recommended practices in RF variables selection analysis;

- GWR calculations (steps 2, 3 and 6), following Gao et al. [91], Farber et al. [93] and Gollini et al. [90] to define kernel type $K_{typ}$ as adaptive, and its function $K_{fun}$ as exponential;

- RF training (steps 7 and 9), following Breiman [41] and Wright et al. [42] for decide default RF calibration (i.e. 500 decision trees, square root number of variables to split at in each node, and permutation method for *LVI* calculation);

- Accuracy assessment (step 12 and 13), calculating Kappa and prediction failure in classification; and R-squared and residual standard error in regression. This assessment is conducted with the OBB samples;

- and Storing outputs (step 14 and 15). These included: *LVI* score for each variable in *Vars*, predictions and probabilities (*YHAT*), and models accuracies (*ACC*).

Since all above calculations were computing demanding, we implemented GWRF for parallel computing but processing time depended of kernel bandwidth $K_{bw}$ (See S1 Appendix). Furthermore, we tested GWRF for classification, reclassifying *DEF* and *REF* rates into 5 classes (see Fig 4), while for regression we maintained them as continuous values. To decide the best approach, we compared results of *ACC* for different values of $K_{bw}$ (see section 1.3.4).

### 1.3.6 DEF and REF linkages assesment

Since *LVI* results were extensive, we first plotted $LVI_{DEF|VARS}$ and $LVI_{REF|VARS}$ in a radar plot [99] to observe how forest change rates were influenced by *Vars*. This facilitated identification of variables with opposite predictive power in *DEF* and *REF*, which were selected to map and observe with more detail. Moreover, we calculated variables correlation with forest change rates to further explore their similitude with *LVI*. For the next step, we followed Freitas et al. [39] and clustered $LVI_{DEF|VARS}$ and $LVI_{REF|VARS}$. For this, we used the expectation-maximization algorithm [100], as it allows continuous and categorical data. We determined 2 clusters based on the gap statistics [101] to later select the one with the highest rate. We assume that these areas represent active forest change fronts with similar variables importance, which distil their driving forces. We named them as $CLUS_{DEF}$ and $CLIS_{REF}$ groups and extracted *Vars* to

test the next hypothesis:

$$H0 : CLUS_{DEF|Vars} = CLUS_{REF|Vars}$$

$$H1 : CLUS_{DEF|Vars} \neq CLUS_{REF|Vars}$$

We applied the Wilcoxon rank sum test [102], which computes P-values that test the $H0$ hypothesis that the two groups have the same distribution. If $H0$ was rejected (P-value > 0.05), we assumed a difference and subtracted the median values to observe if $CLUS_{DEF|Vars}$ was higher or lower than $CLUS_{REF|Vars}$. This operation allows us to classify results for each variable in *Vars* according three categories:

- *DEF* and *REF* were equal ($H0$ is accepted; similar *DEF* and *REF* medians);

- *DEF* was greater ($H0$ is rejected; *DEF* is higher than *REF* median);

- *DEF* was lower ($H0$ is rejected; *DEF* is lower than *REF* median).

Additionally, we calculated the Cliff's Delta [103] to observe the effect size of *Vars* in *DEF* and *REF*. We applied the thresholds provided in Romano [104] to classify this metric into 4 classes (i.e. negligible, small, medium and large) and complete our analysis.

## 1.4 Results

### 1.4.1 Comparison between dasymetric mapping and censuses

We processed 21 census variables from 2001 to 2010 with dasymetric mapping (DAS) to compare their population densities with those derived from unprocessed censuses (CEN). For this, we calculated the population density for each variable and census year, using DAS and CEN data sources to plot them and highlight their differences (Fig 7A). DAS exceeded CEN for variables with larger values (e.g., L_spa, G_pom, G_pof) but fell behind for those with smaller values (e.g., L_wao, H_sma, E_lit). On average, DAS obtained 0.06 ± 0.04 and 0.15 ± 0.12 persons/ha for the population density in 2001 and 2010, while CEN obtained 0.02 ± 0.01 and 0.09 ± 0.07 persons/ha, respectively. This means that DAS exceeded CEN by 145% in 2001 and 58% in 2010. Furthermore, we subtracted population densities in 2001 and 2010 in both sources to obtain their change (Fig 7B). Similarly, it was observed that DAS exceeded CEN for variables with large values (e.g., L_spa, G_pom, E_sec) but was inferior for those with small values (e.g., H_lar, L_wao, E_lit). Averaging all variables, DAS obtained 0.08 ± 0.08 persons/ha for the population density change between 2001 and 2010, while it was 0.07 ± 0.08 persons/ha for CEN; i.e., an increase of 27% by DAS. Interestingly, by DAS, variables H_lar and L_wao showed decreases of –23.2% and –31.2%, respectively, for the population density change between 2001 and 2010.

### 1.4.2 Accuracy assessment of GWRF

We analyzed GWRF according to eight *bw* values and two approaches (classification and regression) to decide which one achieved the highest accuracy. It can be seen that the classification improved the Kappa as the size of *bw* increased, around 400 observations stabilized and variability was reduced. Moreover, results for *REF* were better than for *DEF*, showing in both cases a logarithmic curve with increasing *bw* (Fig 8A). This was shown by different regression results, as the R-squared diminished with *bw* > 100 for *DEF* but was not seen as relevant for *REF* (Fig 8B). This was interpreted as unexpected, as it is know that accuracy increases with larger values of *bw* [93] until its value is large enough to cover all the study area and become a

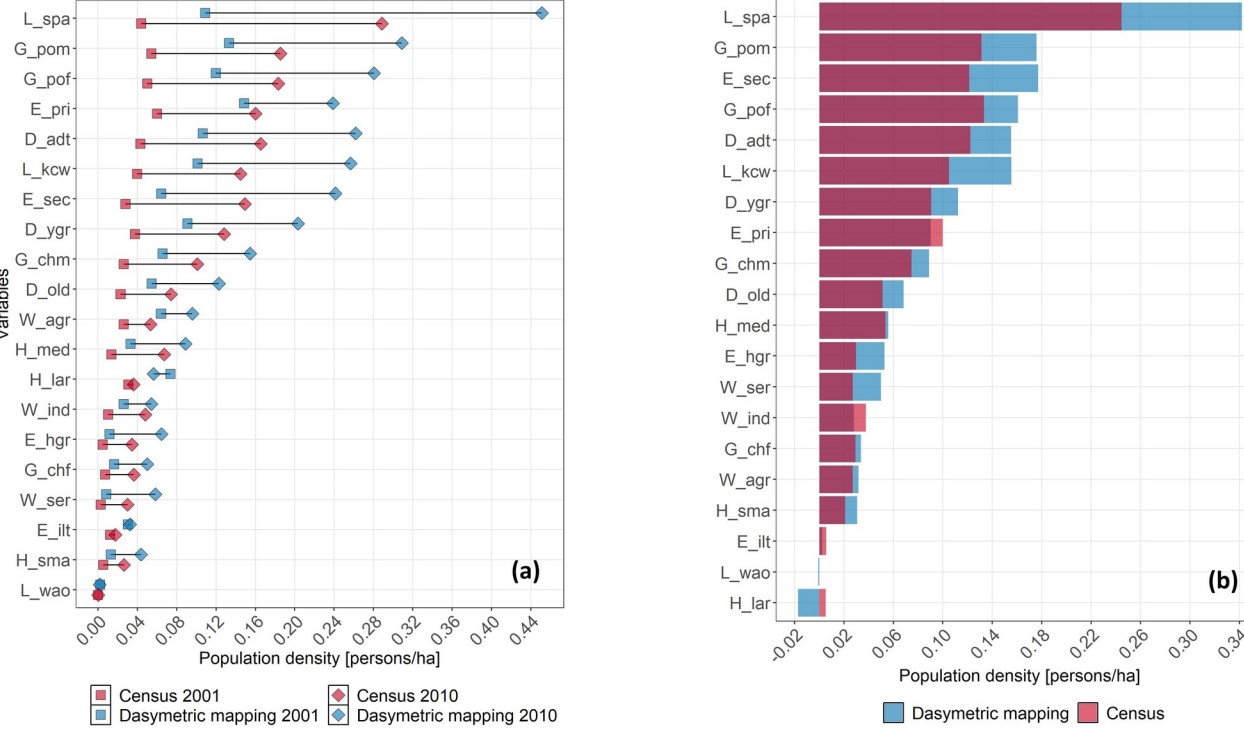

**Fig 7.** (a) Population density differences between dasymetric mapping and census data for 2001 and 2010; and (b) subtraction to derive population density change between 2001 and 2010. In both cases, variables were ordered according to magnitude and transparent color was used to see overlapping areas (red is over blue).

global average. For these reasons, we decided to use a classification approach for *bw* = 400, as larger values did not significantly improve results, achieving a Kappa of 96 ± 2% for *DEF* and 97 ± 1% for *REF*. This indicated that the classification approach resulted in adequate predictions for all classes considered but not in regression. This was probably due to the imbalanced sampling introduced by the kernel during calculations. Finally, we mapped the Kappa for GWRF with *bw* = 400 to observe its spatial distribution (Fig 9). Here we could see that the lowest relative Kappa values (74–95%) covered areas with the largest rates (>2.5%) in *DEF* and *REF*. This implies that GWRF resulted in poor predictions in areas where rates varied (e.g., JS or La Joya de los Sachas in *DEF*, and TN or Tena in *REF*) than in areas with homogeneous rate intervals or where few high rate peaks were observed.

### 1.4.3 LVI comparison and visualization

After the RF classification achieved acceptable results, we created radar plots using the *LVI* results for the four macro levels considered. In the case of Landscape (Fig 10A), it can be seen that variables related to land cover were important for both *DEF* and *REF* but those related to Biophysical seem to be more in *DEF*. Among them, C_pas, C_bls, and B_alt were more important in *REF*, while B_fer was only in *DEF*. Commodities (Fig 10B) indicates that variables related to Infrastructures were more important in *REF* but those related to Agriculture were more important in *DEF*. Among them, variables I_min and I_ngt were more important in *REF*, while A_cao, A_fru and A_plm were more important in *DEF*. The Socioeconomic (Fig 10C) indicate that Work was more important in *REF* but Education was more important in *DEF* with the exception of E_ilt. Furthermore, the variable group Age showed a similar result

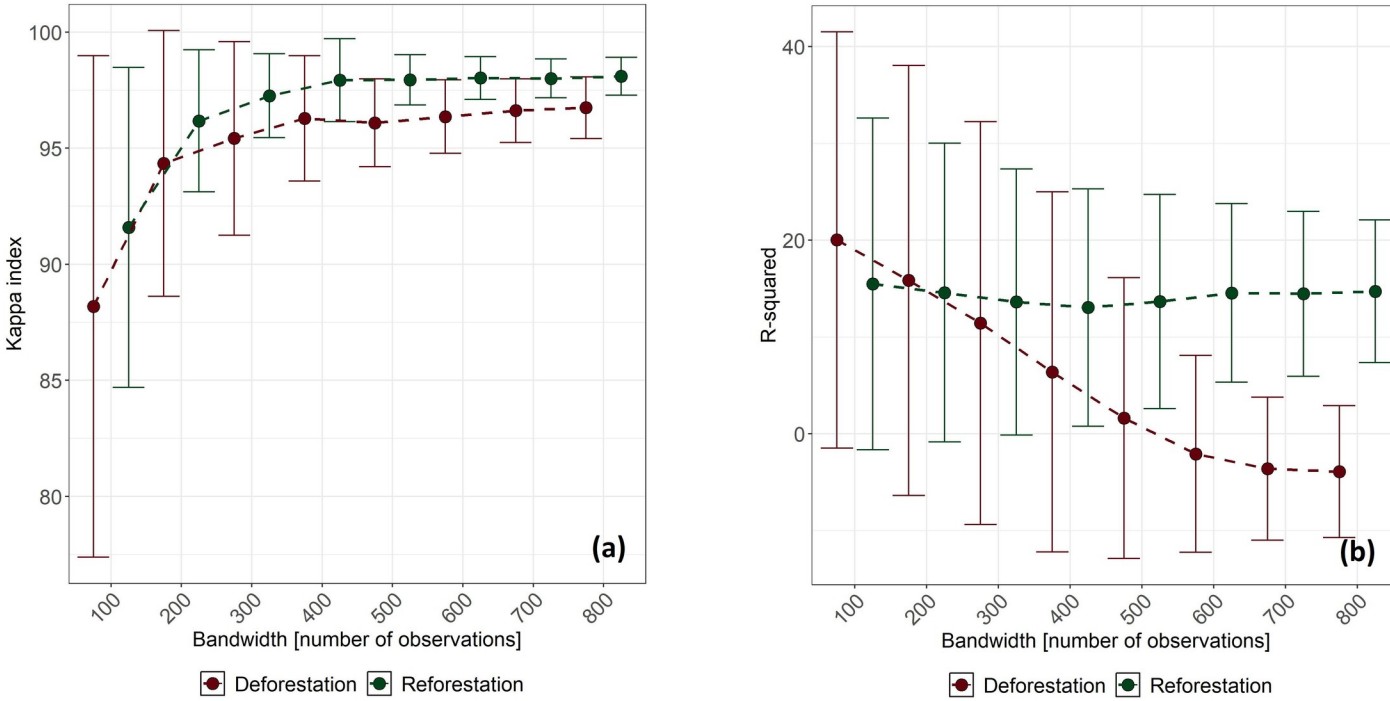

**Fig 8.** Accuracy metrics error bars for different bw sizes in GWRF: (a) classification (kappa) and (b) regression (R-squared). Points connect mean values in Kappa and R-squared, while bars indicate standard deviations.

between *DEF* and *REF*, but variable D_old appears to be more important in *REF*. Finally, the Sociocultural (Fig 10D) indicate that all of them (Gender, Household, and Language) were more important in *DEF*, with a few exceptions. It was seen that G_chf and L_wao variables were only important in *REF*, while the rest of the variables were important in *DEF*.

Following, we mapped variables: B_fer, I_ngt, E_sec and H_med; as they showed opposite predictive power. Fig 11 shows results for *DEF* and here it can be seen that high *LVI* values are spatially related to high *DEF* rates as well. In *REF*, this was different as its observed low to medium *LVI* values (Fig 12) except for variables E_sec and I_ngt whose values are higher in areas with also high *REF* rates. This is particularly interesting, as these variables seem to be good predictors in both *DEF* and *REF* when correlation exists. Following, results from correlations between *LVI* and forest change rates (see S2 Appendix) indicated that E_pri and W_agr were also good predictors (correlation > 0.131), while worse predictors were A_fru and L_wao (correlation < 0.061). It was observed, that the latter were associated to zero *LVI* values (see Fig 10) as this was the result of the cleaning routine of GWRF (see section 1.3.5). Furthermore, it was observed that variables such as G_chm, E_hgr, H_med, L_kcw, B_alt and G_chf generated the opposed effect in *DEF* and *REF*. These variables have inverse relationships and highlights the complex structure of these land cover change dynamics. To facilitate the analysis of these dynamics, in the next section, we report results from the clustering and the hypothesis testing.

### 1.4.4 Clustering and hypothesis testing

After clustering *LVI* into two groups, we tested our hypothesis. In the case of $CLUS_{DEF|Vars}$ we observed that its rate $q$ achieved 2.0 ± 7.5% and included 1160 grid cells (464,000 ha), while in $CLUS_{REF|Vars}$ the rate achieved 4.1 ± 16.4% and included 722 grid cells (288,800 ha). The

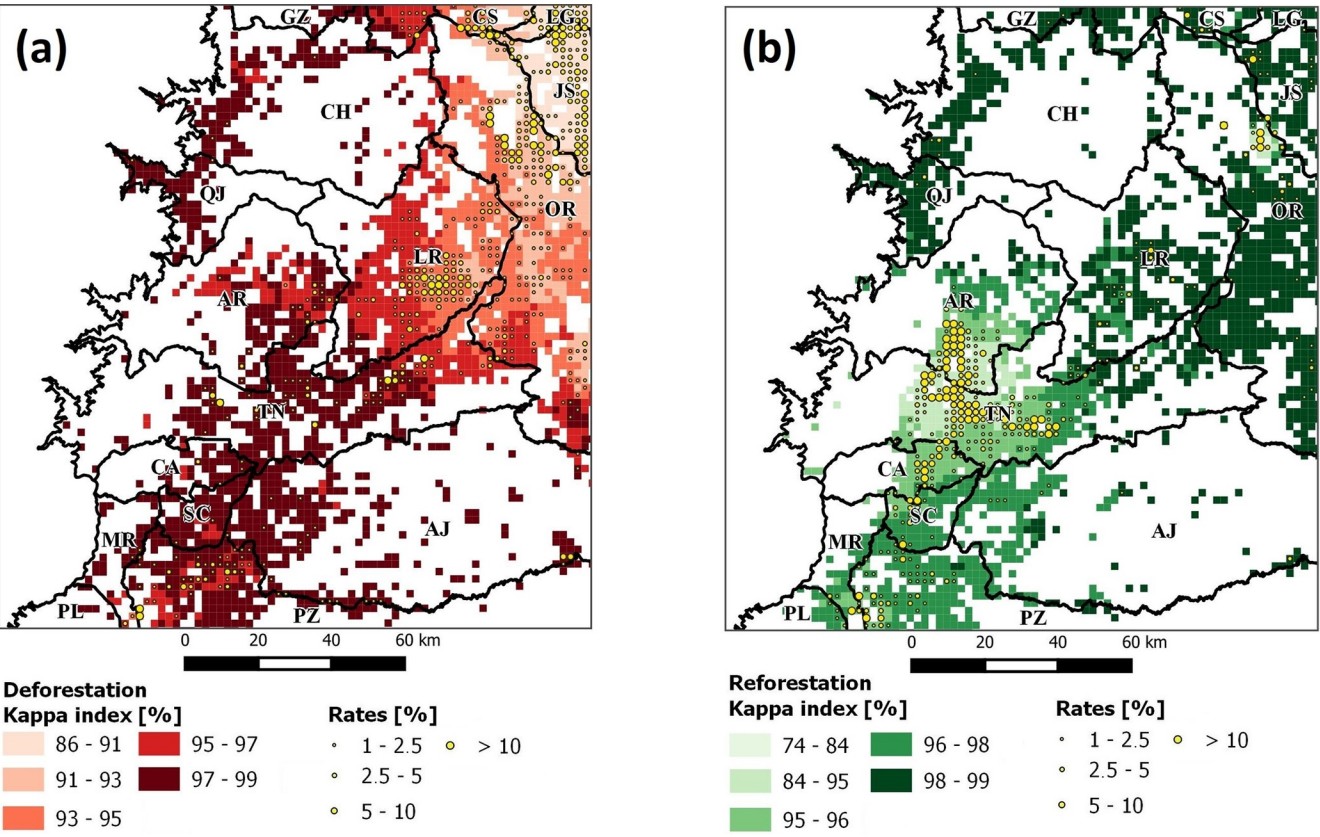

**Fig 9.** Kappa for GWRF classification with b = 400 in (a) *DEF* and (b) *REF* datasets. Rates were filtered to values greater than 1% to enhance visualization.

$CLUS_{DEF|Vars}$ was larger than $CLUS_{REF|Vars}$ with 438 grid cells (175200 ha) but also its rate was lesser by 2.1%. Regarding their locations, both groups matched cantons where higher rates were observed (Fig 13). Interestingly, a boundary between $CLUS_{DEF|Vars}$ and $CLUS_{REF|Vars}$ appears at ~482 m.a.s.l. at cantons CH, LR, TN, and AJ (El Chaco, Loreto, Tena and Arajuno), indicating the limit between these regions and their governing forest change phenomena. Moreover, an overlap was seen between $CLUS_{DEF|Vars}$ and $CLUS_{REF|Vars}$ but it was marginal as it included only three grid cells (or 1200 ha).

Following, we describe the results of the hypothesis. We first show results for variables where $CLUS_{DEF|Vars}$ and $CLUS_{REF|Vars}$ medians were equal. This is summarized in the Table 5 and it can be noted that only variables from the Socioeconomic and Sociocultural macro levels were present. These similitudes were also evidentiated by the Cliff´s delta, as resulting effect sizes were negligible. We identified the next variable groups: Education (E_hgr), Gender (G_chm), Household (H_med, H_sma), Language (L_kcw, L_spa and L_wao) and Work (W_agr) and could observe that variables H_sma (small families) and W_agr (agriculture workers) were close to be significant but their Cliff´s delta indicates that their magnitude effects were still negligible.

The next section belongs to variables whose *H*0 was rejected. We first report variables whose median value was lower in $CLUS_{DEF|Vars}$. This is shown in Table 6 and here its seen variables from Commodities, Landscape and Socioeconomic macro levels exclusively. They were represented by variables groups: Work (W_ind, W_ser), Agriculture (A_mlk, A_cao, A_fru), Infrastructure (I_oil) and Biophysical (B_alt, B_rfl). From them, variables with the largest

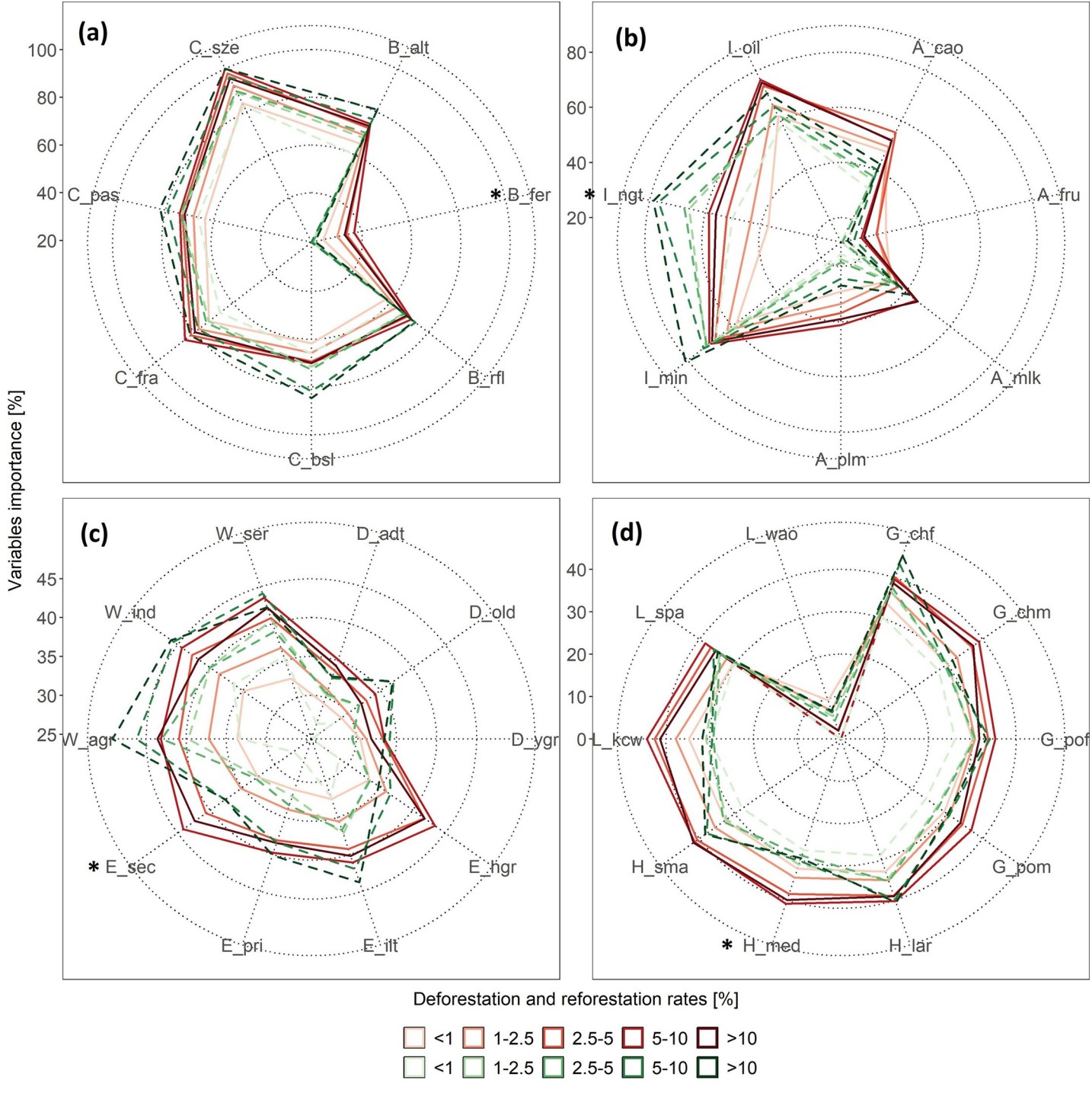

**Fig 10.** LVI radar plot for: (a) landscape, (b) commodities, (c) socioeconomic, and (d) sociocultural macro levels. The asterisk (*) highlight mapped variables.

difference meant that $CLUS_{DEF|Vars}$ was characterized by: higher accessibility to oil palm extraction facilities (A_plm), closer distance to oil wells (I_oil), lower altitudes (B_alt) and lower annual rainfall (B_rfl). Following, higher accessibility to fruit, coffee and cacao collection centers (A_fru and A_cao) was also seen in $CLUS_{DEF|Vars}$ but their differences with $CLUS_{REF|}$

A geographically weighted random forest approach for evaluate forest change drivers in the Ecuadorian Amazon

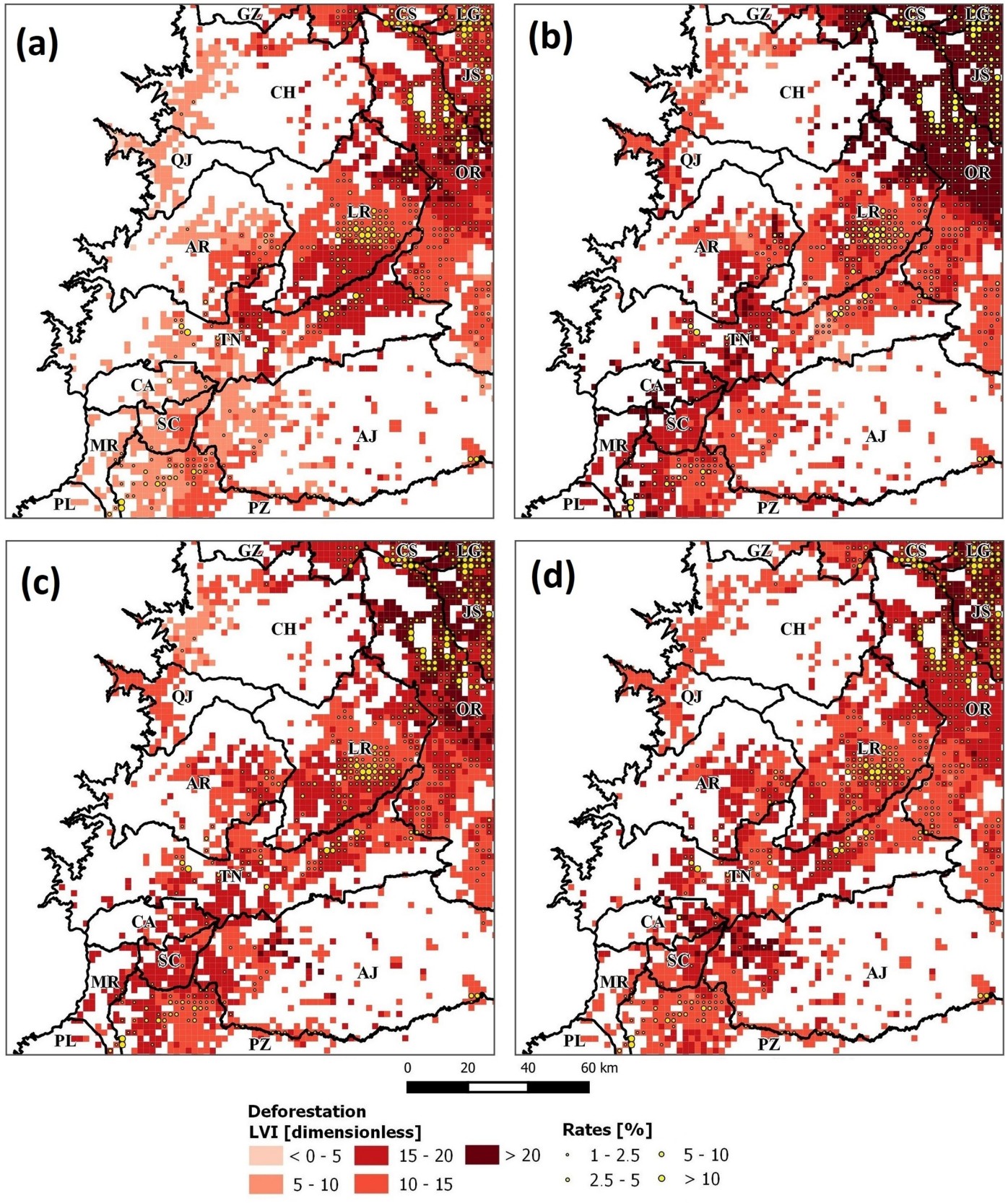

**Fig 11.** $LVI_{DEF|Vars}$ maps for selected variables: a) B_fer (soil fertility), b) I_ngt (stable nighlights), c) E_sec (secondary education); and d) H_med (medium families). High values refer to increased predictive power. Rates were filtered to values greater than 1% to enhance visualization.

$_{Vars}$ were smaller. The opposite picture can be inferred from the abovementioned to describe $CLUS_{REF|Vars}$ characteristics. On the other hand, the zero value of L_wao variable can be attributed to its absence in these regions. Finally, variables A_mlk, W_ind and W_ser seem to achieved negligible differences; therefore, we qualified them as not relevant for our analysis. Contrasting these results, we now report variables whose median value was higher in $CLUS_{DEF|Vars}$. This is shown in Table 7 and in this case, all macro levels were observed. Among identified variable groups, we can mention: Infrastructure (I_ngt, I_min), Biophysical (B_fer), Land cover (C_frac, C_pas, C_sze, C_bsl), Age (D_adt, D_ygr, D_old), Education (E_pri, E_ilt, E_sec), Gender (G_pof, G_pom, G_chf) and Household (H_lar). From them, the only variable that achieved a large difference was bare soil frequency (C_bsl). This indicated that $CLUS_{DEF|Vars}$ was more prone to experience land clearing after tree removal. This is reasonable if we consider that forest succession implies vegetation regrowth after land clearing. Following, variables with medium effect size meant that $CLUS_{DEF|Vars}$ was characterized by: larger patches sizes (C_sze) and higher pasture frequency (C_pas). In addition, other variables with small effect size indicated higher population density for: secondary education (E_sec), older adults (D_old), Illiterate (I_ilt), and female chief household (G_chf). Remaining also in small magnitude, larger distance to mining infrastructures (I_min) seems to also characterize $CLUS_{DEF|Vars}$. For the remaining variables, i.e. I_ngt, B_fer, C_fra, D_adt, D_ygr, E_pri, G_pof, G_pom and H_lar, a negligible effect size is observed and were less informative to identify differences between $CLUS_{DEF|Vars}$ and $CLUS_{REF|Vars}$.

## 1.5 Discussion

### 1.5.1 Utility of remote sensing time series–related products for FDD analysis

Some studies have successfully identified proximate causes in FDD analysis by using remote sensing time series–related products [105,106]. In this research, we extended these applications through the use of (i) grid-based rate calculations, (ii) derivation of land-cover metrics, and (iii) trend analysis of Nighttime Lights Time Series to explore correlations with *DEF* and *REF* rates. Except for the (i), due its simplicity, the other two deserve further discussion as they represent innovative approaches which are not well documented to our current knowledge. Derivation of land-cover metrics indicates that it is possible to extract additional information from remote sensing time series that can be useful for determining the degree of land-use intensity from previous or posterior land-cover change events. This has been done using spectral trajectories [107,108], but here we show how they can be derived from land-cover maps, with a less sophisticated approach and comparatively limited results. With more dense optical and radar time series availability, it might be possible to more precisely detect land-cover classes that are usually not identifiable by their spectral features (e.g., coffee, cacao, forest plantations) but rather by their spectrotemporal signatures, as other studies have demonstrated [109–111]. This could help to improve forest monitoring, but also improve agriculture-related accessibility models, which are more difficult to derive and validate. Furthermore, trend analysis of Nighttime lights Time Series has shown that despite the low spatial resolution, they are still useful for investigating unknown patterns that strengthen model predictions (see Section 1.4.3). This was made possible thanks to free cloud-based platforms that allow processing of vast amounts of data from remote sensing time series and derivation of unprecedented

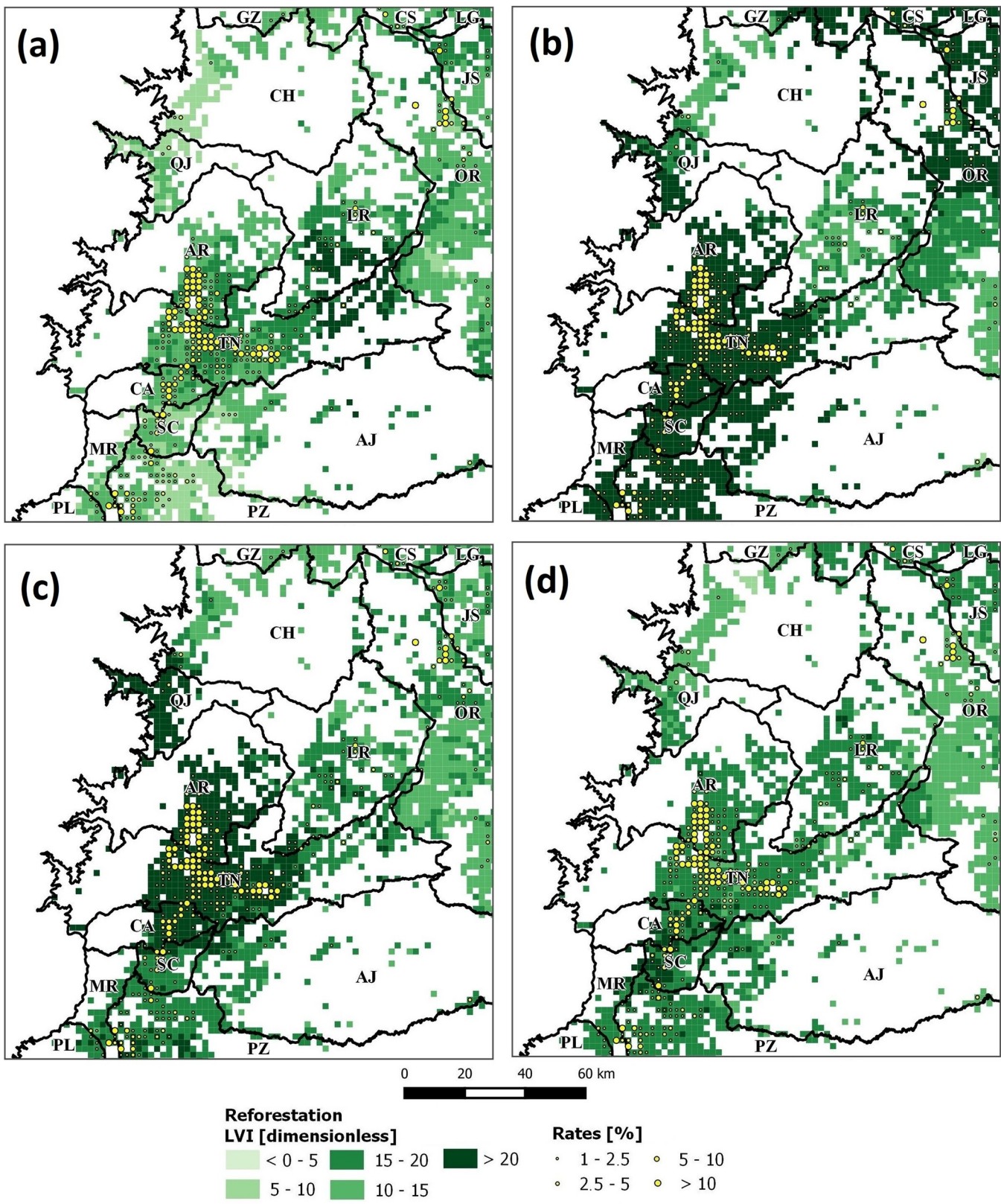

**Fig 12.** $LVI_{REF|Vars}$ maps for selected variables: a) B_fer (soil fertility), b) I_ngt (stable nighlights), c) E_sec (secondary education); and d) H_med (medium families). High values refer to increased predictive power. Rates were filtered to values greater than 1% to enhance visualization.

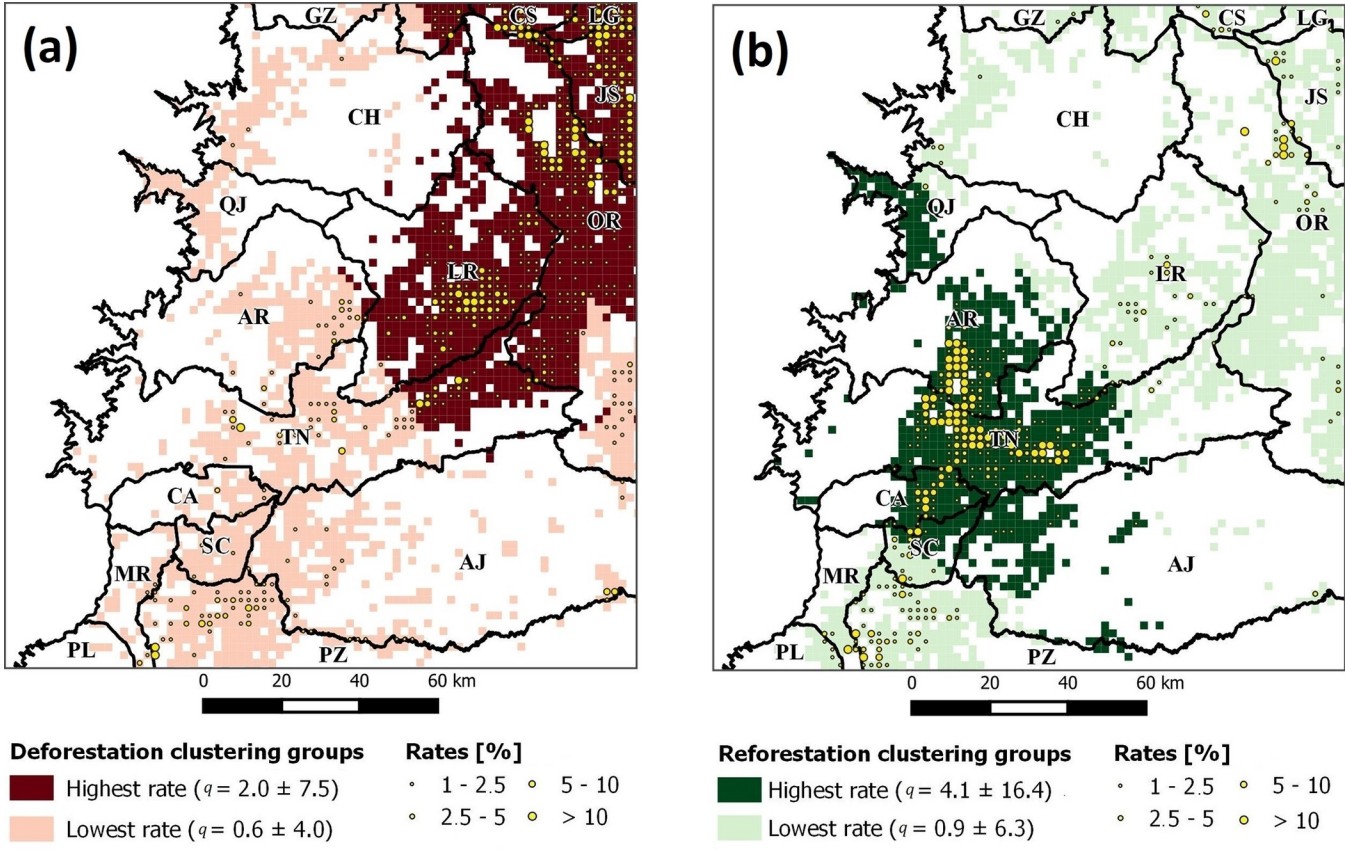

**Fig 13.** Local variable importance (LVI) clustering groups for (a) deforestation (DEF) and (b) reforestation (REF). Rates were filtered to values greater than 1% to enhance visualization.

products [25,112]. This opens new possibilities for future research and reformulating known limitations due to processing capabilities and data availability. This does not mean that field data, novel algorithms, and local knowledge can be shared to such platforms naively, as sensitive information could be exposed and distributed without any ethical concern [113].

**Table 5.** Wilcox test and Cliff´s delta magnitude calculation for variables where *DEF* and *REF* were equal.

| Variable | Median values | | | Wilcox test | | Cliff´s Delta | |
|---|---|---|---|---|---|---|---|
| | *DEF* | *REF* | *DEF–REF* | P–value (*p*) | Significance[1] | Estimate (*d*) | Magnitude[2] |
| E_hgr | 0.0012 | 0.0011 | 0.0001 | 0.2753 | * | 0.0299 | negligible |
| G_chm | 0.0072 | 0.0029 | 0.0042 | 0.3264 | * | 0.0269 | negligible |
| H_med | 0.0021 | 0.0018 | 0.0004 | 0.8309 | * | 0.0058 | negligible |
| H_sma | 0.0018 | 0.0007 | 0.0011 | 0.0507 | ** | 0.0535 | negligible |
| L_kcw | 0.0099 | 0.008 | 0.002 | 0.408 | * | 0.0227 | negligible |
| L_spa | 0.0294 | 0.0264 | 0.003 | 0.5073 | * | -0.0182 | negligible |
| L_wao | 0 | 0 | 0 | 0.1773 | * | -0.0218 | negligible |
| W_agr | 0.0071 | 0.0007 | 0.0064 | 0.0528 | ** | 0.053 | negligible |

[1] Significance thresholds: * ($p > 0.1$), ** ($p < 0.1$), and *** ($p < 0.05$).

[2] Based in Romano 2006: negligible ($d < 0.33$), small ($d < 0.474$), medium ($d < 0.474$), and large ($d > 0.474$).

**Table 6. Wilcox test and Cliff´s delta magnitude calculation for variables where *DEF* is lower.**

| Variable | Median values | | | Wilcox test | | Cliff´s Delta | |
|---|---|---|---|---|---|---|---|
| | *DEF* | *REF* | *DEF–REF* | P-value | Significance[1] | Estimate | Magnitude[2] |
| B_rfl | 3460,5 | 3927,8 | -467,3 | 6.93e-96 | *** | -0,5688 | large |
| B_alt | 332,9 | 639,5 | -306,6 | 1,89e-210 | *** | -0,8475 | large |
| I_oil | 89,1 | 391,6 | -302,5 | 2,61e-195 | *** | -0,8162 | large |
| A_plm | 1–3 | >3 | - | 1,13e-147 | *** | -0,6697 | large |
| A_fru | 1–3 | 1–3 | - | 4,96e-37 | *** | -0,2846 | small |
| A_cao | 0.5–1 | 1–3 | - | 1,86e-33 | *** | -0,3214 | small |
| A_mlk | 1–3 | 1–3 | - | 0,002 | *** | -0,0771 | negligible |
| W_ind | 0,0001 | -0,0005 | -0,0004 | 3,89e-7 | *** | 0,1389 | negligible |
| W_ser | 0,0006 | 0,0004 | 0,0002 | 0,0346 | *** | 0,0578 | negligible |

[1] Significance thresholds: * ($p > 0.1$), ** ($p < 0.1$), and *** ($p < 0.05$).

[2] Based in Romano 2006: negligible ($d < 0.33$), small ($d < 0.474$), medium ($d < 0.474$), and large ($d > 0.474$).

## 1.5.2 DAS achievements and failures in intercensal analysis

Previous studies applying DAS have shown its effectiveness and improved performance for census processing in urban areas [114,115], specifically its capability to harmonize data and allow intercensal analysis. However, few studies have explored DAS in rural areas, as it was done in this study to enhance our comprehension of underlying causes in FDDs. This is because DAS reduces uncertainty in rural population mapping, as census blocks in rural areas are generally large in their extension, depending mostly on larger administrative units and have small population counts. Therefore, their population density calculations result in low

**Table 7. Wilcox test and Cliff´s delta magnitude calculation for variables where *DEF* is greater.**

| Variable | Median values | | | Wilcox test | | Cliff´s Delta | |
|---|---|---|---|---|---|---|---|
| | *DEF* | *REF* | *DEF–REF* | P-value | Significance[1] | Estimate | Magnitude[2] |
| C_bsl | 30,9 | 13,5 | 17,3823 | 2,97e-108 | *** | 0,6035 | large |
| C_sze | 4,7 | 2,4 | 2,3 | 2,096e-43 | *** | 0,3782 | medium |
| C_pas | 81,8 | 71 | 10,8 | 3,13e-56 | *** | 0,4325 | medium |
| G_chf | 0,0017 | 0,0002 | 0,0014 | 1,49e-19 | *** | 0,2476 | small |
| E_sec | 0,0137 | 0,0072 | 0,0065 | 5,25e-9 | *** | 0,1598 | small |
| E_ilt | 0,0002 | -0,0016 | 0,0018 | 2,78e-14 | *** | 0,2083 | small |
| D_old | 0,0068 | 0,0031 | 0,0037 | 2,01e-9 | *** | 0,1642 | small |
| I_min | 88,6 | 60,2 | 28,4 | 2,49e-16 | *** | 0,2244 | small |
| I_ngt | -0,2907 | -0,2906 | -0,0001 | 0,0023 | *** | 0,0709 | negligible |
| B_fer | 1–2 | 1–2 | - | 1,04e-7 | *** | 0,1254 | negligible |
| C_fra | 1,0754 | 1,0707 | 0,0047 | 1,911e-6 | *** | 0,1304 | negligible |
| D_adt | 0,0077 | 0,0024 | 0,0053 | 0,0002 | *** | 0,1011 | negligible |
| D_ygr | 0,006 | -0,0008 | 0,0068 | 7,97e-8 | *** | 0,1469 | negligible |
| E_pri | -0,0008 | -0,003 | 0,0022 | 0,0035 | *** | 0,08 | negligible |
| G_pof | 0,0077 | 0,0016 | 0,0061 | 1,46e-5 | *** | 0,1187 | negligible |
| G_pom | 0,0112 | 0,003 | 0,0082 | 3,73e-7 | *** | 0,1391 | negligible |
| H_lar | -0,0116 | -0,0125 | 0,0009 | 1,63e-5 | *** | 0,118 | negligible |

[1] Significance thresholds: * ($p > 0.1$), ** ($p < 0.1$), and *** ($p < 0.05$).

[2] Based in Romano 2006: negligible ($d < 0.33$), small ($d < 0.474$), medium ($d < 0.474$), and large ($d > 0.474$).

figures that tend to obscure negative trends. Assuming that the data were collected properly, we saw this effect with CEN results, as it hid negative trends for the variables L_wao and H_lar (i.e., variables with small counts), contradicting DAS as well as other research observations in this region [116]. This is particularly important, as future research may consider more precise mapping approaches than choropleths to perform more reliable population density calculations.

Furthermore, as we used a road accessibility model to enhance its location, some observations are worth mentioning. First, this input data incorporated restrictions on non-forest masks with regard to areas less likely to be inhabited. Therefore, their use is valid under the assumption that road accessibility and rural populations are related. However, other transportation sources (e.g., rivers, airfields) may attract rural populations, especially among indigenous groups in the Amazon [117]. This can generate a bias effect that forces allocation of populations to exclusively road-related intervention areas. Moreover, errors in non-forest masks (e.g., confusion with nonanthropic deforestation events, misclassified pixels) could add additional noise that may explain why populations were allocated to areas not known to be occupied (see eastern side of canton CH in Fig 6B, which is a ridge). While identifying and eliminating these artifacts are important tasks in this approach, our recommendation is that future research must improve the methods of rural population mapping before applying DAS, such as using products from Nightlights products with higher spatial resolution than the used in this research.

### 1.5.3 Advantages and limitations of GWR and RF

GWR is a proven methodology for capturing spatial nonstationary relationships, not as a global overview but as a local estimate [118]. However, the use of this approach depends on its calibration (especially for the *bw* parameter) and variable selection to reduce its sensitivity to multicollineary. While some studies have proposed different ways to do this [119,120,121], in this study we present a novel approach using the RF algorithm. Even though it was not possible to determine the impact of variables directly from *LVI*, we could analyze all proposed variables, no matter their multicollinearity, noise, or even type. This represents an advantage in overcoming multicollinearity in GWR, but also selection bias effects [122], which are more complex to control in multivariate problems. Moreover, LVI and its mapping showed the predictive power of selected variables that helped not only to identify those relevant for modelling but also their spatial extent. This subsequently facilitated to extract regions with similar physical and human impact characteristics, which allowed us to discuss with more detail where spatial determinants were relevant to *DEF* and *REF*. This strengthens the need for better strategies in land planning. Also future work is needed to explore more applications of GWRFC, as we do not discuss other additional results (e.g. prediction probabilities) or RF model interpretation approaches (e.g. partial dependence plots), which are also possible to derive using this methodology (See S3 Appendix). Furthermore, clustering of *LVI* spatial representations to later extract the impact of variables without any transformation of original values should be considered as another advantage. While Wilcoxon rank sum test allowed to identify a similitude or difference between rates and variables; it was the Cliff´s Delta test, which gave additional detail to quantify these findings. Critics of null hypothesis significance testing [123, 124] and GWR [125], may have found this procedure more convenient that application of parametric approaches in GWR, as assumptions failures and bias effects are less relevant to non-parametric algorithms such as RF. Its further *LVI* clustering and identification of focus areas allowed to conduct exploratory analysis of original data and statistical tests to better discriminate specific variables interactions.

Nevertheless, some limitations of the proposed methodology are important to mention. Our approach does not constitute a GWR but rather a geographically weighted RF classification (GWRFC; see S3 Appendix). As seen in Section 1.4.2, RF regression achieved a relatively poor performance with respect to GWRFC, forcing our FDD analysis from a regression to a classification problem. That is why we discretized our independent variables into classes and up-sampled unbalanced cases during RF training. The latter operation allowed the GWRFC to obtain acceptable results, as other studies also found [126]; however, is unknown whether unbalanced sampling affected RF regression as well. Breiman [41] also warned the limited performance of RF regression that may be also applied to our results. This highlights the need for further research, and our recommendation is that experimentation with other nonparametric algorithms, especially for regression analysis (e.g., support vector machines, neural networks) should be considered, as novel studies has shown [127]. However, we must caution that GWR is more accepted as an exploratory or interpolation technique rather than a predictive tool [128,129], something already known in the literature but with only marginal discussion [130].

### 1.5.4 Linkages between DEF and REF in the NEA

Prior works on FDD analysis in Latin America have documented contrasting dynamics where population growth, socioeconomic development, and agricultural expansion affect *DEF* and *REF* differently [131]. In our research, we extend these findings to identify more specific and localized FDDs, which enrich the explanations from those already known in the NEA (see Section 1.1). With respect to their location, our results indicated two hotspots that highlighted the tendency for *DEF* and *REF* to be spatially clustered, which supports Fagua et al. [132], showing that forest change is not an accidental process, but rather is determined by geographical location and intensity. In this regard, *DEF* showed a relationship with intense land-use changes associated with oil extraction, increasing nightlight intensity, suitability for commercial agriculture [133], and accesibility to facilities (especially for palm oil, coffee and cacao). This landscape verifies the expansion of the oil industry, economic oportunities, and colonization of the northern and western Ecuadorian Amazon [21,57,134]. This is in contrast to *REF*, as its biophysical setting (>482 m.a.n.m.) indicated less suitability for commercial agriculture (except for coffeee and cacao) and diminished accesibility to their facilities. Moreover, an increasing distance from oil wells but less distance from mining blocks indicates other natural resource extraction interests [135]. Here, agroforestry systems with patches less than 4 ha combining secondary forests, cacao, and coffee plantations seem to dominate the landscape. This is similar to the traditional "chakra" land-use system described by Torres et al. [136] and to naturally regenerated forests as a consequence of the abandonment of degraded pastures due to nutritional limitations of soils in the region [137,138]. The latter may explain why accessibility to milk production facilities was better than other related agriculture products in *REF*, but also compares well with Rudel et al. [9] for the highest probability of *REF* at the shortest distances to roads. This was manifested especially in abandoned pastures where *colonos* experienced an important out-migration from the late 1980s, as is also described by Carr [81] as a regional trend in Latin America. Nevertheless, future research may consider incorporate migration censuses to corroborate these findings and identify where they manifest locally.

These differences between *DEF* and *REF* landscapes were also reflected in their demographic structure. Despite people of all ages (especially older adults, i.e. 45–72 y) from *colonos* and *Kichwas* groups related to agricultural activities and secondary education showing more of a link to *DEF* than *REF*, we found some variables that highlighted their particularities. In this respect, the diminishing trend of high fertility and large families, which favors more *REF* than *DEF*, is remarkable. This resembles the demographic and forest transition theories

[11,139] that fit well considering the economic development after the discovery of oil in the Ecuadorian Amazon. Another finding in this direction is increasing education years that seems to have a positive effect on *DEF* and *REF* but the latter only when is related to higher education. This suggests environmental externalities produced by education, as has also been reported in other regions [140,141]. Furthermore, a slight *DEF* association was seen where the male population exceeded the female population, a phenomenon observed in other studies [142,143], but it was not true at all in our case, as chief female households were more strongly associated with *DEF*, similar to the results of Sellers [144]. However, we found that the number of chief male households exceeded female households for both *DEF* and *REF* indicated different proportions. This suggests that land-tenure and land-use decision-making is mostly dominated by males in the NEA, but this could be different among ethnicities, since a diminished effect in both *DEF* and *REF* was observed for the *Kichwa* group respect to *colonos* but little or no effect compared with the *Huaorani* group. This agrees with the results of Sierra et al. [145], who reported levels of *DEF* and *REF* of 42.7% and 35.7% in the *Kichwa* territory and 0.3% and 0.4% in the *Huaorani* territory for almost the same period of time (2000–2008). This indicates that language (as a proxy of ethnicity) together with gender should be considered in future research to better characterize and discuss FDDs, as other studies have also suggested [146,147].

## 1.6 Conclusions

This research underlines the importance of downscaling global problems to the local scale and assessing individual drivers of land-use change in coupled socioecological systems. Applying an experimental methodology fusing remote sensing time series products, dasymetric mapping, and GWRFC, we were able to support the analysis of the spatial distribution of the population and forest dynamics in the Ecuadorian Amazon in more detail. Our findings reveal that at the local scale, key FDDs identified at the global scale can be better described. This was demonstrated in our study, as different groups played different roles in forest change, with varying impact in different regions in the NEA. Accessibility to agricultural collection centers and distance to infrastructure had an influence on both *DEF* and *REF*. However, biophysical and land-cover variable groups demonstrated that they could not be minimized, since they are ancillary sources that support and corroborate findings focused on them, i.e., describing suitable conditions for agriculture or natural resource extraction. Furthermore, socioeconomic and sociocultural variable groups had a strong influence on untangling population dynamics and their relationship with forest change, which made interpreting the results challenging and final statements fuzzier. Nevertheless, combining forest dynamics and population information in a geospatial environment underlines their variable complexity and extent. Combining aspects of livelihood patterns can be more meaningful than using proxies to represent individual aspects. The results of this study also highlight the roles of education, gender, and language in forest dynamics, which are more studied in social sciences but therefore show a strong relevance also for environmental studies. Interdisciplinary expertise and transdisciplinary exchange are needed to foster a better understanding of coupled socioecological systems from local to global scales. This can only be facilitated by inter- and transdisciplinary research.

## Supporting information

**S1 Appendix. Processing time for multiple bandwidth sizes using a sample dataset of 1000 obs. with 34 variables.**
(JPG)

**S2 Appendix. Correlations between *LVI*, *DEF* and *REF* rates.**
(JPG)

**S3 Appendix. Source code of GWRFC algorithm.** https://github.com/FSantosCodes/
GWRFC.
(DOCX)

## Acknowledgments

We want to thanks to our colleagues from the Center for Development Research (ZFL), Center for Remote Sensing of Land Surfaces (ZEF) and Remote Sensing Research Group (RSRG) in Bonn-Germany. We also thanks to collegues from the AI Saturdays-Quito, Universidad Tecnológica Indoamérica, and Ministerio del Ambiente in Quito-Ecuador for their comments during this research. Special thanks to the Secretaría de Educación Superior, Ciencia, Tecnología e Innovación (SENESCYT) and Universidad Tecnológica Indoamérica for funding this research. The authors also thanks to R, OSGeo, USGS and Google Earth Engine communities for sharing their knowledge and provide no-cost tools and data. Thanks also to two anonymous reviewers and José Jara for their comments on an earlier version of this paper.

## Author Contributions

**Conceptualization:** Fabián Santos, Valerie Graw.

**Data curation:** Fabián Santos.

**Formal analysis:** Fabián Santos.

**Funding acquisition:** Fabián Santos, Santiago Bonilla.

**Investigation:** Fabián Santos.

**Methodology:** Fabián Santos, Valerie Graw, Santiago Bonilla.

**Project administration:** Fabián Santos.

**Resources:** Fabián Santos, Valerie Graw, Santiago Bonilla.

**Software:** Fabián Santos.

**Supervision:** Fabián Santos, Valerie Graw.

**Validation:** Fabián Santos, Valerie Graw, Santiago Bonilla.

**Visualization:** Fabián Santos.

**Writing – original draft:** Fabián Santos.

**Writing – review & editing:** Valerie Graw, Santiago Bonilla.

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
