## [Decision Letter · Decision Letter 0]

3 Sep 2019

PONE-D-19-19468

Evaluating Forest Change Drivers in the Northern Ecuadorian Amazon: A Geographically Weighted Random Forest Approach

PLOS ONE

Dear Mr. Santos,

Thank you for submitting your manuscript to PLOS ONE. After careful consideration, we feel that it has merit but does not fully meet PLOS ONE’s publication criteria as it currently stands. Therefore, we invite you to submit a revised version of the manuscript that addresses the points raised during the review process.

We would appreciate receiving your revised manuscript by Oct 18 2019 11:59PM. To enhance the reproducibility of your results, we recommend that if applicable you deposit your laboratory protocols in protocols.io, where a protocol can be assigned its own identifier (DOI) such that it can be cited independently in the future. For instructions see: http://journals.plos.org/plosone/s/submission-guidelines#loc-laboratory-protocols

We look forward to receiving your revised manuscript.

Kind regards,

Zhenlong Li, Ph.D.

Academic Editor

PLOS ONE

Journal Requirements:

2.  We note that Figure(s) in your submission contain [map/satellite] images which may be copyrighted. All PLOS content is published under the Creative Commons Attribution License (CC BY 4.0), which means that the manuscript, images, and Supporting Information files will be freely available online, and any third party is permitted to access, download, copy, distribute, and use these materials in any way, even commercially, with proper attribution. For these reasons, we cannot publish previously copyrighted maps or satellite images created using proprietary data, such as Google software (Google Maps, Street View, and Earth). For more information, see our copyright guidelines: http://journals.plos.org/plosone/s/licenses-and-copyright.

1.    You may seek permission from the original copyright holder of Figure(s) [#] to publish the content specifically under the CC BY 4.0 license. 

3. We note you have included a table to which you do not refer in the text of your manuscript. Please ensure that you refer to Table 4 in your text; if accepted, production will need this reference to link the reader to the Table.

Additional Editor Comments (if provided):

I now have received the review comments from the two reviewers. As you see, one reviewer is fully satisfied with the revised paper. The other reviewer, however, still has some major concerns about the method and clarification the manuscript. Therefore, I invite you to conduct another round of revision to address/respond to the second reviewer's concerns.

Reviewers' comments:

Reviewer's Responses to Questions

**Comments to the Author**

1. Is the manuscript technically sound, and do the data support the conclusions?

Reviewer #1: Yes

Reviewer #2: Partly

2. Has the statistical analysis been performed appropriately and rigorously? 

Reviewer #1: Yes

Reviewer #2: No

3. Have the authors made all data underlying the findings in their manuscript fully available?

Reviewer #1: Yes

Reviewer #2: Yes

4. Is the manuscript presented in an intelligible fashion and written in standard English?

Reviewer #1: Yes

Reviewer #2: Yes

5. Review Comments to the Author

Reviewer #1: Authors did a great job in handling my previous questions. In this version, it is a lot more clear about the overall workflow and function of each step. I recommend publishing this paper.

Reviewer #2: I still think that this paper discussed a good idea as it tried to combine deforestation and reforestation, global and local driving factors, as well as social and ecological variables. However, this revision version is totally different with the previous one as you changed the methodology from GWR ridge regression into GWR-RF. Honestly, I think the previous version is better, you just need to make an improvement based on reviewer’s comment. Regarding this version, there are some issues that need more clarification:

1. You should be able to distinguish between spatial autocorrelation and multicollinearity. Did you use RF to overcome spatial autocorrelation or multicollinearity? Spatial autocorrelation is correlation between samples or data, whereas multicollinearity is correlation between independent variables.

2. I don’t understand what is the role of GWR here? I don’t see any result of GWR, except you only use it to calculate b (bandwith) value. Even I don’t see any GWR equation, what is the dependent variable and what are the independent variables? How to define the relationship between GWR and RF? All these questions need to be clarified.

3. You show the R-squared for accuracy assessment as can be seen in Figure 9B as a result of RF regression. Since the result is very low, then you decline RF regression then use RF classification. How did you combine the results of RF classification with GWR? According to figure 3 you used GWR+RF to get high accuracy of LVI, but how you did it? It needs an explanation

4. Another accuracy assessment that you used is Kappa Index. It is not clear how do you get the Kappa index (Fig 9A)? I mean what is the base map that you used to get the Kappa value? As you know the Kappa index is calculated based on the comparison/overlay between a reference map and other maps as a a result of the analysis.

5. Suddenly you used radar plots to map the LVI (local variable importance) according to deforestation and reforestation. This is never been mentioned in the methods section or Figure 3. What is the relationship between RF classification and radar plot here? As far as I know RF classification is a strong instrument for determining the importance of variables in classification.

6. What is the connection between radar plot and LVI clustering? I see both showed the relationship between variables and deforestation or reforestation.

7. In LVI clustering, you only made two clusters with high rate and low rate deforestation or reforestation, then you use Wilcoxon test to define the similarity between variables that found in different cluster of deforestation and reforestation. This is also missing from the method section. It needs an explanation.

8. As you only use RF classification and not regression you cannot measure the impact of LVI over forest changes rates as you mentioned in Figure 3. You are only able to map the correlation between LVI and forest changes. Need more evidence to prove that this correlation indicating causal relationship.

9. Still, I feel this paper put more emphasize on the model than explaining the underlying reasons of your findings. I think you need to sharpen the discussion and conclusion section.

6. PLOS authors have the option to publish the peer review history of their article (what does this mean?). If published, this will include your full peer review and any attached files.

Reviewer #1: No

Reviewer #2: Yes: Didit Okta Pribadi

---

## [Author Response · Author response to Decision Letter 0]

19 Nov 2019

Dear Editor,

We have prepared a revised version of the manuscript, which adress comments from last review. Please refer to the cover letter for more details.

In behalf of the authors,

Dr. Santos.

---

## [Editor Report · Decision Letter 1]

22 Nov 2019

A Geographically Weighted Random Forest Approach for evaluate Forest Change Drivers in the Northern Ecuadorian Amazon

PONE-D-19-19468R1

Dear Dr. Santos,

We are pleased to inform you that your manuscript has been judged scientifically suitable for publication and will be formally accepted for publication once it complies with all outstanding technical requirements.

With kind regards,

Zhenlong Li, Ph.D.

Academic Editor

PLOS ONE
---

## [Editor Report · Acceptance letter]

5 Dec 2019

PONE-D-19-19468R1 

A Geographically Weighted Random Forest Approach for evaluate Forest Change Drivers in the Northern Ecuadorian Amazon 

Dear Dr. Santos:

I am pleased to inform you that your manuscript has been deemed suitable for publication in PLOS ONE. Congratulations! Your manuscript is now with our production department. 

With kind regards,

on behalf of

Dr. Zhenlong Li 

Academic Editor

PLOS ONE